# Transcriptome-wide association analysis of brain structures yields insights into pleiotropy with complex neuropsychiatric traits

Bingxin Zhao[1,8], Yue Shan[1,8], Yue Yang [1], Zhaolong Yu [2], Tengfei Li[3,4], Xifeng Wang[1], Tianyou Luo[1], Ziliang Zhu[1], Patrick Sullivan[5], Hongyu Zhao [2,6], Yun Li [1,5,7,9✉] & Hongtu Zhu [1,4,9✉]

Structural variations of the human brain are heritable and highly polygenic traits, with hundreds of associated genes identified in recent genome-wide association studies (GWAS). Transcriptome-wide association studies (TWAS) can both prioritize these GWAS findings and also identify additional gene-trait associations. Here we perform cross-tissue TWAS analysis of 211 structural neuroimaging and discover 278 associated genes exceeding Bonferroni significance threshold of $1.04 \times 10^{-8}$. The TWAS-significant genes for brain structures have been linked to a wide range of complex traits in different domains. Through TWAS gene-based polygenic risk scores (PRS) prediction, we find that TWAS PRS gains substantial power in association analysis compared to conventional variant-based GWAS PRS, and up to 6.97% of phenotypic variance (p-value $= 7.56 \times 10^{-31}$) can be explained in independent testing data sets. In conclusion, our study illustrates that TWAS can be a powerful supplement to traditional GWAS in imaging genetics studies for gene discovery-validation, genetic co-architecture analysis, and polygenic risk prediction.

[1] Department of Biostatistics, University of North Carolina at Chapel Hill, Chapel Hill, NC, USA. [2] Interdepartmental Program in Computational Biology and Bioinformatics, Yale University, New Haven, CT, USA. [3] Department of Radiology, University of North Carolina at Chapel Hill, Chapel Hill, NC, USA. [4] Biomedical Research Imaging Center, School of Medicine, University of North Carolina at Chapel Hill, Chapel Hill, NC, USA. [5] Department of Genetics, University of North Carolina at Chapel Hill, Chapel Hill, NC, USA. [6] Department of Biostatistics, Yale University, New Haven, CT, USA. [7] Department of Computer Science, University of North Carolina at Chapel Hill, Chapel Hill, NC, USA. [8] These authors contributed equally: Bingxin Zhao, Yue Shan. [9] These authors jointly supervised this work: Yun Li, Hongtu Zhu. ✉email: yunli@med.unc.edu; htzhu@email.unc.edu

Variations in brain structure and microstructure across individuals are associated with many neurological and psychiatric (referred to as neuropsychiatric hereafter) traits including cognitive functions[1–5], neurodegenerative, neurodevelopmental, and psychiatric disorders[6–9], as well as alcohol and tobacco consumption[10], and physical bone density[11]. Structural variations of human brain can be quantified by multimodal magnetic resonance imaging (MRI). Specifically, the T1-weighted MRI (T1-MRI) can provide basic morphometric information of brain tissues, such as volume, surface area, sulcal depth, and cortical thickness. In region of interest (ROI)-based T1-MRI analysis, images are annotated onto ROIs of pre-defined brain atlas, and then both global (e.g., whole brain, gray matter, white matter) and local (e.g., basal ganglia structures, limbic, and diencephalic regions) markers can be generated to measure the brain anatomy. On the other hand, diffusion MRI (dMRI) can capture local tissue microstructure through the random movement of water. Using diffusion tensor imaging (DTI) models, brain structural connectivity can be quantified by using white matter tracts extracted from dMRI, which build psychical connections among brain ROIs and are involved in connected networks for various brain functions[12,13]. See Miller et al.[11] and Elliott et al.[14] for a global overview and more information about neuroimaging modalities used in the present study.

Structural neuroimaging traits have shown moderate-to-high degree of heritability in both twin and population-based studies[14–24]. In the past decade, genome-wide association studies (GWAS)[14,24–34] have been conducted to identify the associated genetic variants (typically single-nucleotide polymorphisms [SNPs]) for brain structures. A highly polygenic[35,36] genetic architecture has been observed, indicating that a large number of genetic variants contribute to variations in brain structure measured by neuroimaging biomarkers[21,37]. Particularly, using data from the UK Biobank (UKB[38]) cohort, two recent large-scale GWAS have identified 578 associated genes for 101 regional brain volumes derived from T1-MRI[39] (referred to as ROI volumes, $n = 19,629$) and 110 DTI parameters of dMRI[40] (referred as DTI parameters, $n = 17,706$). Some of these discovered genes had been implicated for neuropsychiatric diseases or traits by previous GWAS. However, most of them have not been verified and need further investigations. Complementary to traditional GWAS, transcriptome-wide association studies (TWAS) have become increasingly adopted in gene-trait association analysis thanks to recent advances in gene expression imputation methods[41–47] and burgeoning generation of such expression imputation reference data sets (e.g., the Genotype-Tissue Expression (GTEx) project[48]). Despite some challenges[49] such as interpreting causality, TWAS has successfully discovered additional gene-trait associations and provided insights into biological mechanisms for many complex traits[50]. Through imputed transcriptomes, TWAS can reduce the multiple testing burden and leverage gene expression data to increase testing power for gene-trait association detection. This is a particularly desirable feature for imaging genetics studies, for which most neuroimaging GWAS data sets continue to have small sample sizes and heavy multiple testing burden[51].

In this work, we performed TWAS analysis for 211 structural neuroimaging traits including 101 ROI volumes and 110 DTI parameters. As these brain-related traits tend to be highly polygenic[21,37] and are related to many traits across a range of categories[11], we used a cross-tissue (panel) TWAS approach (UTMOST[43]) in our main analysis. UTMOST first performs single-tissue gene-trait association analysis in each reference panel with both within-tissue and cross-tissue statistical penalties, and then combines these single-tissue results using the Generalized Berk-Jones (GBJ) test[52], which accommodates tissue dependence and can account for the potential sharing of local expression regulation across tissues. The UKB data set was used in the discovery phase ($n = 19,629$ for ROI volumes and 17,706 for DTI parameters, respectively). For the discovery UKB cohort, we compared TWAS-significant genes with previous GWAS findings in gene-based association analysis via MAGMA[53] and gene-level functional mapping and annotation results by FUMA[54]. The UKB TWAS results were validated in five independent data sources, including Philadelphia Neurodevelopmental Cohort (PNC[55], $n = 537$), Alzheimer's Disease Neuroimaging Initiative (ADNI[56], $n = 860$), Pediatric Imaging, Neurocognition, and Genetics (PING[57], $n = 461$), the Human Connectome Project (HCP[58], $n = 334$), and the ENIGMA2[24] and ENIGMA-CHARGE collaboration[34] ($n = 13,193$, for eight ROI volume traits, referred as ENIGMA in this paper). Chromatin interaction enrichment analysis was conducted for TWAS-significant genes. Finally, we developed TWAS gene-based polygenic risk scores[59] (PRS) using FUSION[41] to fully assess polygenic architecture and examine the predictive capability of the UKB TWAS results.

## Results

**Overview of TWAS discovery-validation in the six data sets.** We conducted a two-phase discovery-validation TWAS analysis for 211 neuroimaging traits by using the UKB cohort for discovery and the other data sets (ADNI, HCP, PING, PNC, and ENIGMA) for validation. We applied the UTMOST gene expression imputation models trained on GTEx tissues, and used GWAS summary statistics generated from previous GWAS as inputs. We refer to $1.04 \times 10^{-8}$ (that is, $5 \times 10^{-2}/22,694/211$, adjusted for all candidate genes and traits performed) as the significance threshold for gene-trait associations unless otherwise stated. The original version of UTMOST models was trained using GTEx v6 as the reference. In this study, we retrained the UTMOST models using the recently released GTEx v8 data and performed our analysis using both versions. As the GTEx v6 and v8 databases share individual-level samples, we are particularly interested in the associations that can be consistently detected in the two versions. Therefore, in the rest of this paper we reported genes that were either (1) significant in both versions; or (2) significant in one version and were within ±1 MB window with at least one significant gene in the other version (Methods).

The UKB discovery phase identified 918 significant gene-trait associations (Supplementary Data 1) between 278 genes and 152 neuroimaging traits (57 ROI volumes, 95 DTI parameters). Of the 278 TWAS-significant genes, 90 (32.4%) had significant associations with more than two neuroimaging traits, 16 (10.4%) had more than five significant associations, and 16 (5.8%) had at least ten, including *POLR2F, TREH OR1F12, FOXF1, LRRC37A, AC008105.1, MAPT, ARHGAP27, EIF4EBP3, PLEKHM1, ZKSCAN4, CCDC157, XRCC4, AC005670.1, CRHR1*, and *RECQL4*. These 16 genes together contributed 344 (37.5%) of the 918 gene-trait associations, indicating their widespread influences on brain structures. Specifically, we identified 173 genes whose imputed gene expression levels were significantly associated with one or more of the 57 ROI volumes (328 associations in total, 186 additional, Supplementary Fig. 1), and 140 significantly associated genes (35 overlappings) for one or more of the 95 DTI parameters (590 associations in total, 277 additional, Supplementary Fig. 2).

Figure 1 illustrates that TWAS prioritized previous GWAS findings of MAGMA and FUMA and also discovered many additional associations and genes. Moreover, some genes were associated with both ROI volumes and DTI parameters, while others were more specifically related to certain structures

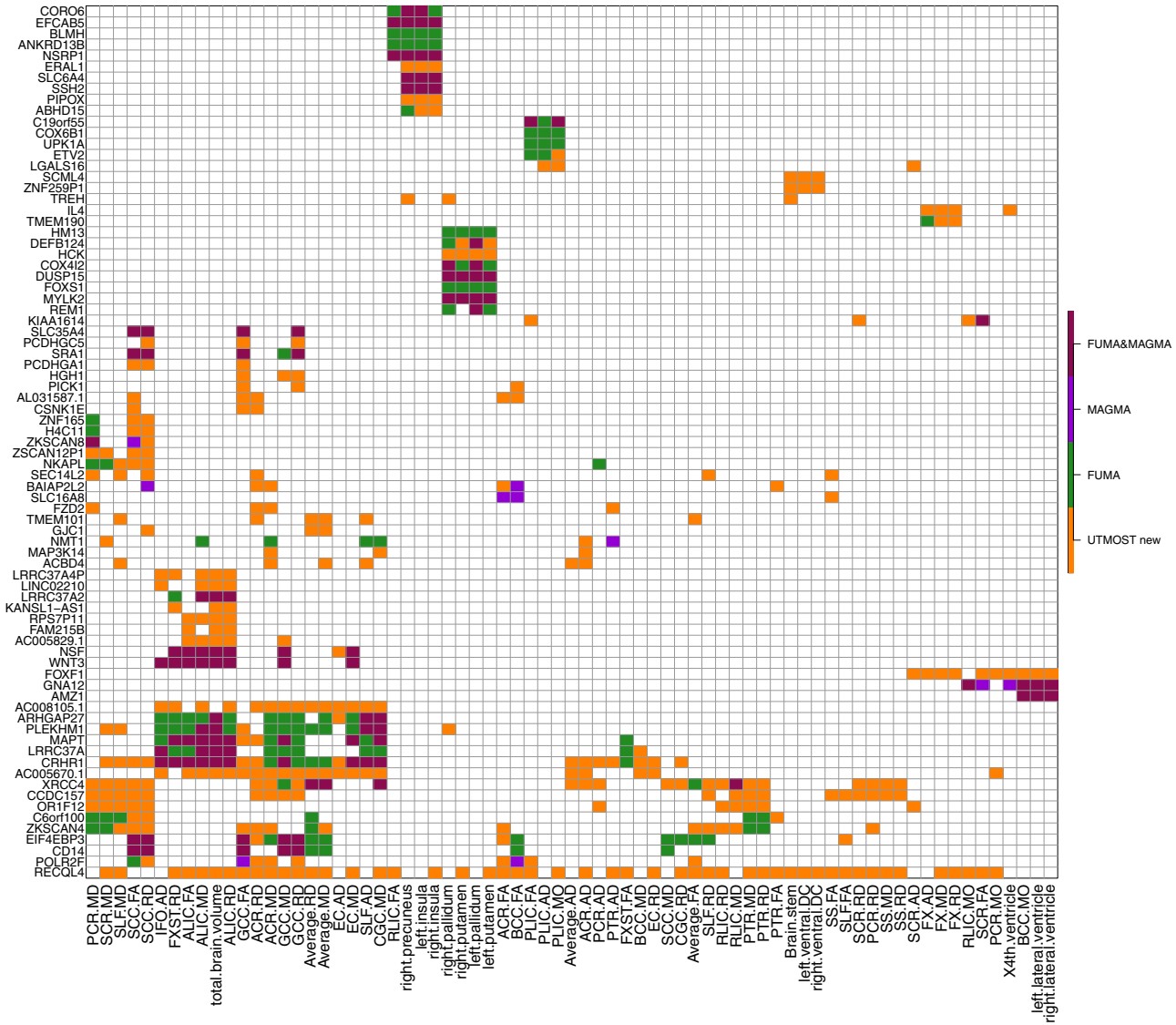

**Fig. 1 Selected significant gene-trait associations discovered in UKB (UK Biobank) cross-tissue TWAS analysis of 211 neuroimaging traits (*n* = 19,629 subjects for ROI volumes and 17,706 for DTI parameters).** The gene-level associations were estimated and tested by the cross-tissue UTMOST approach (https://github.com/Joker-Jerome/UTMOST). We used the *p* value threshold of $1.04 \times 10^{-8}$, corresponding to adjusting for testing 211 imaging phenotypes with the Bonferroni correction. The *x* axis provides the IDs of the neuroimaging traits, and the *y* axis lists the detected genes in TWAS. The additional (UTMOST new) and previously reported GWAS-significant associations (MAGMA, FUMA, and FUMA&MAGMA) were labeled with different colors (orange, purple, green, and red, respectively).

(Supplementary Fig. 3). For example, *XRCC4, ZKSCAN4, EIF4EBP3*, and *CD14* were associated with DTI parameters but not ROI volumes, *DEFB124, COX4I2, HCK, HM13*, and *REM1* showed associations with putamen and pallidum volumes, and the associations of *PLEKHM1, LRRC37A, MAPT, AC005670.1, RECQL4, ARHGAP27*, and *CRHR1* were spread widely across DTI parameters and total brain volume.

We validated the UKB results in the other five independent cohorts. For each data set, we applied the Bonferroni-corrected significance threshold accounting for all candidate genes and traits analyzed (that is, $5 \times 10^{-2}/22,694/$number of traits, Supplementary Data 2–6). We found that 19 additional UKB TWAS-significant genes (*NPSR1, TREH, CRYBA1, MFRP, SLX1B, RPL13AP3, GALP, KCNH7, DCTPP1, LINC02454, JPH3, IL4, HCK, TIMM8AP1, LGALS3, LINC02057, RECQL4, DLGAP5*, and *AC090666.1*) can be validated in one or more of the five data sets. These data sets also replicated six previous UKB GWAS-significant genes (*NUP210L, MIR1-1HG, DOK5,*

*KRTAP5-1, AC008393.1*, and *DPP4*), and four genes that were significant in both UKB TWAS and GWAS (*DCC, LRRC37A, ANKRD42*, and *DLG2*) (Supplementary Fig. 4). The TWAS additional findings and validated genes were discussed further in detail below.

**Additional TWAS discoveries and validated genes.** Of the 278 UKB TWAS-significant genes, 159 were not discovered in previous GWAS of the same UKB data set (Supplementary Data 7). TWAS resulted in 102 additional associated genes for 54 ROI volumes (186 associations, Supplementary Fig. 5), and 75 additional genes for 90 DTI parameters (277 associations, Supplementary Fig. 6). According to NHGRI-EBI GWAS catalog[60], the 159 TWAS-significant genes replicated 21 previous findings on brain structures, including *JPH3*[61] for hippocampal volume in mild cognitive impairment, *CRYBA1*[33] for brain stem volume measurement, *AC145285.2*[33] for caudate nucleus volume, and *C1QL1*[62] for white matter hyperintensity burden. The other 138

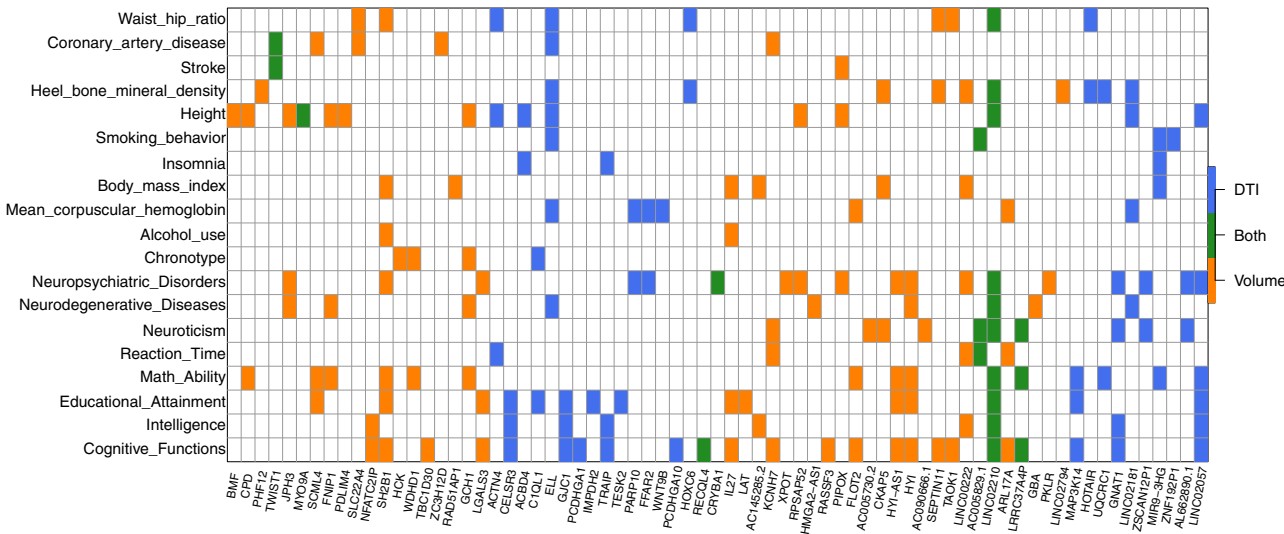

**Fig. 2 Cross-tissue TWAS-significant genes of neuroimaging traits (***n* **= 19,629 subjects for ROI volumes and 17,706 for DTI parameters) that have been linked to other complex traits in previous GWAS.** For each of the cross-tissue TWAS-significant genes listed in the *x* axis, we manually checked the previously reported associations on the NHGRI-EBI GWAS catalog (https://www.ebi.ac.uk/gwas/). The genes associated with DTI parameters (DTI), ROI volumes (volume), and both of them (Both) were labeled with three different colors (blue, orange, and green, respectively).

genes had not been linked to brain structure previously and thus can be regarded as additional genes for these 211 neuroimaging traits. To explore the genetic overlaps with other traits in different domains, we performed association lookups for the 159 TWAS genes on the NHGRI-EBI GWAS catalog. Figure 2 shows that these genes were widely associated with anthropometric measures (e.g., height, waist-to-hip ratio, heel bone mineral density, body mass index), neuropsychiatric traits (e.g., cognitive function, intelligence, math ability, schizophrenia, bipolar disorder, Alzheimer's disease), coronary artery disease, mean corpuscular hemoglobin, neuroticism, education, reaction time, chronotype, smoking behavior, and alcohol use, such as *ELL*[63–65], *SH2B1*[66–69], *IL27*[68,70], *KCNH7*[71,72], *HYI*[73,74], and *GNAT1*[75,76].

For the 29 TWAS-validated genes shown in Supplementary Fig. 4, ten (*ANKRD42, DCC, LRRC37A, NUP210L, DOK5, KRTAP5-1, MIR1-1HG, AC008393.1, DLG2,* and *DPP4*) of them had been discovered in the previous UKB GWAS and were implicated in brain-related complex traits, such as neuroticism[77], major depression[78], schizophrenia[75,79,80], Intelligence[70], math ability[72], reaction time[68], and insomnia[81]. The remaining 19 genes, which are additional findings from our TWAS analysis, also had known associations with various neuropsychiatric traits. For example, previous GWAS reported that *HCK* was associated with chronotype[81], *LGALS3* with schizophrenia[82], *AC090666.1* with neuroticism[71], *CRYBA1* with depression[78], *RECQL4* with cognitive ability[68], *KCNH7* with cognitive performance[72] and reaction time[68], and *JPH3* with bipolar disorder[83] and cognitive impairment[61]. Moreover, we found that *DCC, MIR1-1HG, DPP4,* and *RECQL4* were specifically associated with brain-related traits and disorders, while other genes (such as *NUP210L, DLG2, AC090666.1, KCNH7,* and *JPH3*) were also widely associated with non-brain traits, including triglycerides[84], mean platelet volume[64], and coronary artery disease[85]. In summary, TWAS additional and validated genes expand the overview of gene-level pleiotropy across these traits, suggesting that neuroimaging-derived biomarkers could be useful in studying a wide range of complex traits.

**Comparing power to detect the association between brain tissues and all tissues.** As a comparison, we performed a brain tissue-specific version of UTMOST TWAS that only combined

brain tissues (10 brain tissues in GTEx v6 or 13 brain tissues in GTEx v8, Method). This brain tissue-specific TWAS detected 396 significant gene-trait associations (Supplementary Data 8) between 134 unique genes and 81 neuroimaging traits, including 84 associated genes for one or more of 29 ROI volumes (136 associations, Supplementary Fig. 7), and 68 genes (18 overlapping) for one or more of 52 DTI parameters (260 associations, Supplementary Fig. 8).

Most (119/134) of the brain tissue-specific genes have been identified by either the cross-tissue TWAS (117/134) or previous GWAS (65/134). The 15 genes that were uniquely identified by brain tissue-specific analysis included *DNAJC2, LHFPL3, NUPR1, UQCRQ, BCL2L1, MBD2, KNCN, NUFIP2, MIB2, C3orf62, CDHR4, FXYD1, TMEM173, ZSCAN31,* and *PI4KAP2*. Among them, *LHFPL3* showed associations with education[86], social behavior[87,88], cognitive ability[68], schizophrenia[89], and bipolar disorder[90]. *MBD2* was associated with reaction time[68], *ZSCAN31* with schizophrenia[89] and cross disorders[91], and *NUPR1, CDHR4,* and *C3orf62* with intelligence[81,92].

Compared with brain tissue-specific TWAS, the cross-tissue analysis clearly identified more signals. For example, of the 328 gene-trait associations identified by cross-tissue analysis of ROI volumes, 142 had been identified in GWAS, 50 can be additionally identified by brain tissue-specific TWAS, and 136 can only be detected by cross-tissue analysis (Supplementary Fig. 9). Similarly, 313 of the 590 cross-tissue TWAS associations for DTI can be identified in GWAS, 90 can be additionally identified by brain tissue-specific TWAS, and 187 were cross-tissue TWAS only (Supplementary Fig. 10). These results illustrate the advantage of cross-tissue analysis over brain tissue-specific TWAS for discovering association signals that are difficult to be identified in traditional GWAS. We further compared their results in a few follow-up analyses below.

**Comparison with GWAS variant-level signals and conditional analysis.** For each of the 918 gene-trait associations detected in cross-tissue TWAS, we used previous GWAS summary statistics to check the most significant variant within the gene region (with a 1 MB window on each side) that was pinpointed in the same UKB data set (Method). The GWAS *p* value of the most significant variant (i.e., the variant with the smallest *p* value) was

$>1 \times 10^{-6}$ for associations of 19 genes (Supplementary Data 9). None of them had been identified by MAGMA or FUMA, indicating that it can be difficult to detect these genes by GWAS or post-GWAS screening for any of these neuroimaging traits. Of the 19 genes, seven (*GALP, LINC02057, CRYBA1 TREH, IL4, DCTPP1, RECQL4*) were validated in one or more of the five validation data sets and were discussed in the previous section. For the other 12 genes (*LGALS16, MYO9A, FAM83C, CEACAMP3, H4C11, AC005670.1, OR10V3P, TMEM136, CELSR3, TMEM101, CCDC157,* and *GDF5*) genes, *MYO9A* was reported for defects in the structure and function of the neuromuscular junction[93], *FAM83* family was linked to certain brain tumors[94], *CELSR3* was associated with education[71] and cognitive ability[70,77], and *CCDC157* was found to be associated with white matter microstructure in other data sets[95]. The same checking was then performed for the 396 significant gene-trait associations of brain tissue-specific TWAS. We found that only *DCTPP1* and *CCDC157* had minimum GWAS $p$ value $<1 \times 10^{-6}$ (Supplementary Data 10).

We next performed a conditional analysis to see whether the TWAS signals remained significant after adjustment for the most significant genetic variant used in UTMOST gene expression imputation models (Method). Although our cross-tissue analysis combined information from many genetic variants across various human tissues, we found that 472 associations may indeed be dominated by the strongest GWAS signal of the imputation model, as their conditional p-values were larger than 0.05 (Supplementary Data 11). However, the conditional $p$ values of eight genes (*WIF1, XRCC4, C15orf56, CCDC53, RPSAP52, CCDC157, AMZ1, NMT1*) were smaller than $1 \times 10^{-6}$ for 23 gene-trait associations, suggesting that these associations were unlikely to be driven by a signal genetic variant. When the $p$ value threshold was relaxed to $1 \times 10^{-3}$, 118 associations of 42 genes persisted after conditional analysis. Similar conditional analysis was also performed on significant associations of brain tissue-specific TWAS. The conditional $p$ values were smaller than $1 \times 10^{-6}$ for five genes (*XRCC4, C15orf56, NMT1, CCDC157, AMZ1*) with 20 associations, and were smaller than $1 \times 10^{-3}$ for 25 genes with 84 associations (Supplementary Data 12).

**Chromatin interaction enrichment and genetic overlaps**. To explore the biological interpretations of TWAS and GWAS-significant genes, we performed enrichment analysis in promoter-related chromatin interactions of four types of brain cells[96] (induced pluripotent stem cells (iPSC)-induced excitatory neurons, iPSC-derived hippocampal DG-like neurons, iPSC-induced lower motor neurons, and primary astrocytes) (Method). Both GWAS and cross-tissue TWAS-significant genes were significantly enriched in chromatin interactions of astrocytic glial cells (Supplementary Data 13, Wilcoxon rank test, $p$ value $< 2.8 \times 10^{-2}$), and combining GWAS and cross-tissue TWAS-significant genes resulted in a smaller p value ($1.04 \times 10^{-3}$). Cross-tissue TWAS-significant genes were also significantly enriched in chromatin interactions from two neuron types (excitatory and lower motor neurons). For all of the three neuron types, cross-tissue TWAS-significant genes had smaller enrichment $p$ values ($p$ value range = $[2.3 \times 10^{-2}, 6.18 \times 10^{-2}]$) than those of GWAS-significant genes ($p$ value range = $[0.11, 0.57]$). Overall, these results suggest that cross-tissue TWAS-significant genes were more actively interacted with other chromatin regions and may play a more important role in regulating gene expressions as compared with other genes. In contrast, brain tissue-specific TWAS-significant genes did not show any significant enrichment ($p$ value range = $[0.14, 0.68]$), indicating the value of cross-tissue TWAS over brain-tissue-specific TWAS.

Next, we applied fastENLOC[97] to perform colocalization analysis for the 278 cross-tissue TWAS-significant genes (Methods). We found that 96 of the 278 (34.5%) genes (involving 233 of 918 gene-trait associations) had regional colocalization probability (RCP) > 0.1 in at least one tissue type and seven genes (involving 17 gene-trait associations) had RCP > 0.9 (Supplementary Data 14). Among them, there are known risk genes. For example, *SLC16A8* is a known risk gene of glioma/glioblastomas[98]. In our cross-tissue TWAS analysis, *SLC16A8* was significantly associated with multiple white matter microstructure traits, and fastENLOC colocalization analysis also found that *SLC16A8* had a high colocalization probability (0.919) with expression quantitative trait loci (eQTL) signals in GTEx v8 nerve tibial tissue type.

To further explore the gene-level genetic overlaps among brain structure and other complex traits and clinical outcomes, we performed cross-tissue TWAS analysis for 16 other brain-related complex traits with a large GWAS sample size, including neuropsychiatric traits, cognition, and cardiovascular risk factors (Supplementary Data 15). We found that 112 of the 278 cross-tissue TWAS-significant genes of neuroimaging traits were also significantly associated with one or more of 14 traits (that is, $5 \times 10^{-2}/22,694/16$, Supplementary Data 16, Fig. 3). These results suggest the genes involved in brain structure changes are often related to vascular risk factors and are also active in brain functions and neuropsychiatric disorder/diseases. For example, we found 65 overlapping genes with cognitive function, 54 with education, 53 with numerical reasoning, 50 with intelligence, 39 with neuroticism, 37 with drinking behavior, and 22 with schizophrenia. A large proportion (83/112) of these genes were associated with more than one neuropsychiatric traits, and 13 genes were linked to more than five traits, including *NSF, LRP4, ZSCAN9, CRHR1, ARHGAP27, RECQL4, C1QTNF4, KCNH7, MAPT, FAM180B, AC005829.1, AC005670.1,* and *AC090666.1,* indicating the high degree of statistical pleiotropy[99] of these genes.

We next performed some additional analysis for the 19 validated UKB TWAS additional genes. First, we found that *JPH3* has a high probability of being loss-of-function (LoF) intolerant[100] (pLI = 0.986), indicating its intolerant of LoF variation. *JPH3* has also been reported for brain disorders, including Huntington disease[101,102], Huntington Disease-Like 2[101,103], spinocerebellar ataxia[101], and Dentatorubral-pallidoluysian atrophy[104]. Second, *DCTPP1* and *DLGAP5* were also identified by a recent eQTL study of developing human brain[105]. Moreover, *LGALS3* and *DLGAP5* were within the mitotic progenitors and cell division function module in the constructed transcriptional networks[106], and *JPH3* was within the adult neurons, synaptic transmission, and neuron projection development function module, indicating their potential functions in biological processes of brain development. In addition, *NPSR1, GALP, KCNH7, JPH3, IL4,* and *LGALS3* mutations have been reported to be related with behavior/neurological phenotypes in mice (Mouse Genome Informatics, http://www.informatics.jax.org/).

**TWAS gene-based polygenic risk scores analysis**. To fully assess the polygenic genetic architecture of neuroimaging traits and examine the predictive ability of UKB TWAS results, we constructed TWAS gene-based PRS on subjects in PNC, HCP, PING, and ADNI cohorts for all of the 211 neuroimaging traits (Method). The prediction analysis was conducted separately on 52 reference panels (13 GETx v7 brain tissues, 35 GTEx v7 other tissues, 1 non-GETx brain tissue, and 3 non-GETx other tissues) using the FUSION[41] software and database. We found that genetically predicted profiles for 28 ROI volumes (Fig. 4) and 23 DTI parameters (Supplementary Fig. 11) were significantly associated with the corresponding observed traits in all testing

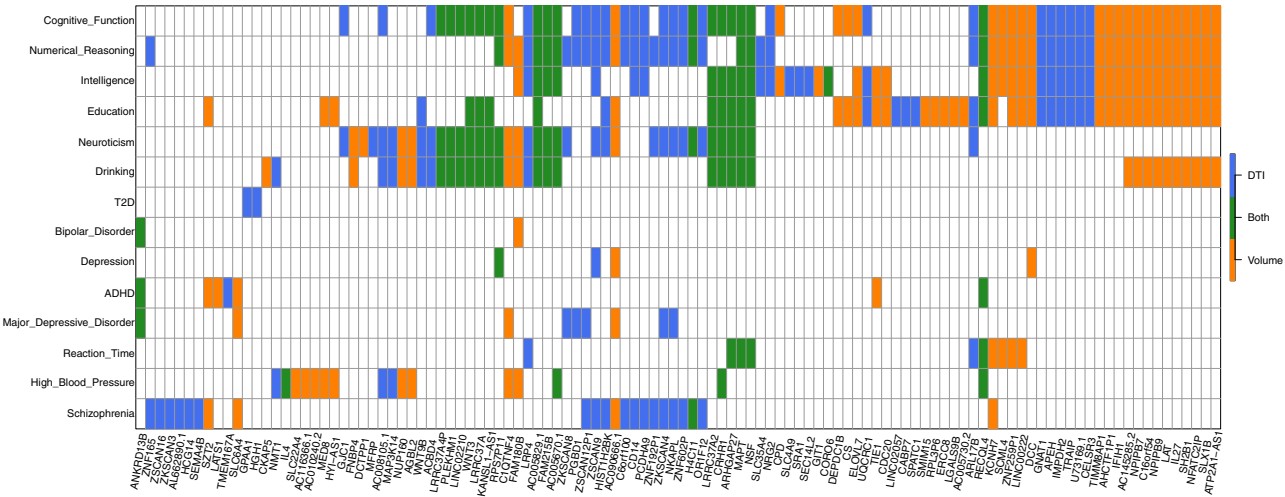

**Fig. 3 Overlapping cross-tissue TWAS-significant genes between neuroimaging traits ($n = 19,629$ subjects for ROI volumes and 17,706 for DTI parameters) and other complex traits and clinical outcomes.** The gene-level associations were estimated and tested by the cross-tissue UTMOST approach (https://github.com/Joker-Jerome/UTMOST). We adjusted for testing 211 neuroimaging traits ($p$ value threshold $1.04 \times 10^{-8}$) and 16 other traits ($p$ value threshold $1.37 \times 10^{-7}$) with the Bonferroni correction, respectively. The $x$ axis provides the IDs of the neuroimaging traits. The $y$ axis lists the 16 other traits, and Supplementary Data 15 details the resources of their GWAS summary statistics and the sample sizes of corresponding studies.

data sets after Bonferroni correction (that is, $101 \times 4 + 3 \times 110 = 734$ tests). Compared with previous SNP-based PRS analysis that yielded significant PRS profiles for 11 ROI volumes[39], gene-based PRS profiles were significant for more ROI volumes, such as left/right insula, left/right pallidum, left/right ventral DC, left/right fusiform, and left/right transverse temporal, suggesting the substantial power gain in association analysis of PRS. The significant TWAS PRS can account for 0.97–6.97% phenotypic variance ($p$ value range $= [8.0 \times 10^{-29}, 6.81 \times 10^{-5}]$) (Supplementary Data 17–18), which was within a similar range to SNP-based PRS analysis (1.17–6.38%)[39]. For example, the (incremental) $R^2$ of TWAS PRS of cerebellar vermal lobules VIII–X was 6.97% in PNC and 6.48% in HCP, and the $R^2$ of SFO MD-derived TWAS PRS was 3.8% in PING and 2.41% in PNC.

To evaluate the additional prediction power that TWAS PRS has on the top of traditional GWAS PRS, we next include both GWAS and TWAS PRS together as predictors in one linear model to predict the above 28 TWAS-significant ROI volumes (Method). Compared to the linear model with TWAS or GWAS PRS only, we found that the prediction accuracy was improved for most ROIs when using both of the two types of PRS (Fig. 5). Conditioning on GWAS PRS, TWAS PRS can additionally explain 0.33–5.22% of phenotypic variance (Supplementary Data 19, Supplementary Fig. 12). The two PRS together can have 1.48–9.02% prediction $R^2$ (Supplementary Data 20, Supplementary Fig. 13). For example, the $R^2$ of cerebellar vermal lobules VIII–X became 7.94% in PNC and 9.02% in HCP, in which TWAS PRS additionally contributed 5.22% and 3.66% for PNC and HCP, respectively. On the other hand, conditioning on TWAS PRS, GWAS PRS increased the $R^2$ by 0.02–4.65% (Supplementary Data 21, Supplementary Fig. 14). These results clearly demonstrate the unique value of TWAS PRS for complex traits prediction and suggest that combining both GWAS and TWAS PRS can achieve better prediction accuracy.

We also examined the performance of each reference panel on these significant traits. There was a significant linear relationship between the panel sample size and average prediction $R^2$ (48 GTEx reference panels, simple correlation $= 0.53$, $p$ value $= 1.21 \times 10^{-4}$, Supplementary Fig. 15), which means that currently, the panel sample size may dominate the performance of TWAS PRS analysis regardless of the tissue specificity[59]. Among the

brain tissue panels, we found that cerebellum tissue had the largest sample size and also showed the highest average $R^2$ (Supplementary Data 22), further supporting the importance of reference panel sample size. Thus, we expect that a reference panel with a larger sample size will be available and can improve the prediction power of TWAS PRS.

**Discussion**

In this study, we applied TWAS methods on 211 neuroimaging traits to identify genes, whose imputed expression levels were associated with brain structure variations. Using a cross-tissue approach, our main discovery analysis identified 138 additional genes and validated 29 significant genes at stringent Bonferroni correction $p$ value thresholds. Conditional analysis and comparison with GWAS variant-level results suggested that the identification and validation of additional genes reflect the ability of TWAS to reduce the testing burden and to combine the small genetic variant effects. We also performed brain tissue-specific TWAS and illustrated the unique strengths of cross-tissue TWAS in conditional and enrichment analyses. Lots of brain structure-related genes were known genetic factors for a wide range of complex traits, ranging from physical traits, cognition, mental disease/disorders, blood assays, to lifestyle, which extend the potential applications of neuroimaging traits. Some of these genetic overlaps were additionally highlighted by a TWAS analysis of other complex traits.

The present study faces some limitations. First, as these results are purely based on statistical associations, it is hard to draw conclusions about the underlying causality and prioritize causal genes[43,107]. This is also one of the main challenges for most of the current TWAS approaches[49]. Follow-up experimental validation is a clear need to confirm TWAS results and pinpoint the causal genes of brain structure changes. In addition, colocalization analysis (such as fastENLOC) can also help prioritize genes having more evidence of causal association. Second, the brain tissue-specific TWAS did not yield much additional results compared with the previous GWAS, and brain tissue panels did not show better prediction accuracy than non-brain tissues in gene-based PRS analysis. Both of the two observations support the use of multiple tissues in our analysis to increase testing power for association analysis, but making the causality

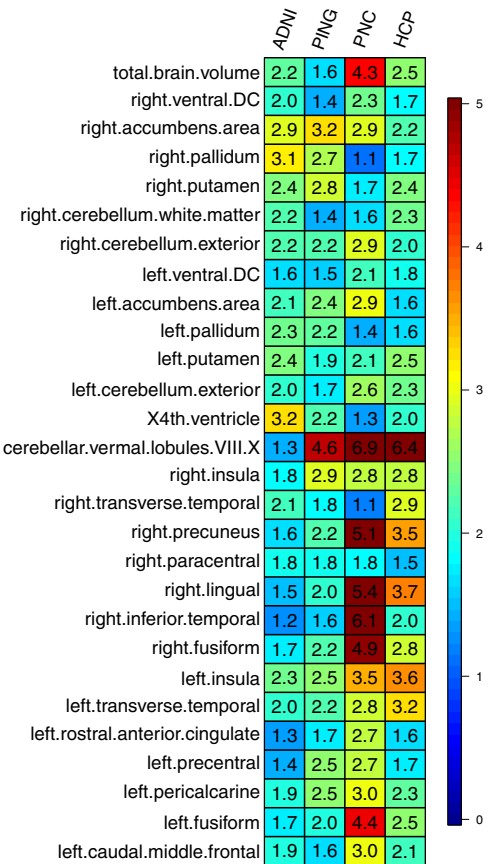

**Fig. 4 Prediction accuracy (incremental $R^2$) of gene-based polygenic risk scores constructed by UKB TWAS results ($n = 19,629$ subjects) on the four independent data sets.** The x axis lists the four independent cohorts (ADNI, HCP, PING, and PNC) and the y axis lists the ROI volumes. The displayed numbers are the proportions of phenotypic variation that can be additionally explained by UKB TWAS-derived gene-based PRS.

interpretation of TWAS results even more complicated. The better performance of cross-tissue analysis may be partially explained by the fact that multi-tissue approaches additionally evaluate cross-tissue evidence[108,109]. In addition, though gene-based PRS had much better power in association tests than SNP-based polygenic scores, their prediction accuracies were similar. These limitations may be due to the fact that current brain tissue reference panels, like many other tissues, do not have large sample sizes, and/or the associated gene expression imputations may be of low quality. For example, imputations using genetic variants with low frequency may not be accurate when the reference panel sample size is small. Despite these limitations, TWAS has been holding and delivering to the promise of becoming a powerful supplement to traditional GWAS in imaging genetics studies. In our study, many additional gene-trait associations were discovered and the underlying genetic overlaps among complex traits were substantially expanded. With better brain tissue gene expression reference panels and more neuroimaging GWAS data sets available, future TWAS analyses of neuroimaging traits are expected to show the value of tissue specificity and improve our understanding of the genetic basis of human brain.

## Methods

**GWAS summary statistics data sets**. We made use of GWAS summary statistics to test for gene-trait associations in our TWAS study. The GWAS summary-level were from six studies, including the UKB[38] (http://www.ukbiobank.ac.uk/resources/)

study, the HCP[58] (https://www.humanconnectome.org/) study, the PING[57] (http://www.chd.ucsd.edu/research/ping-study.html) study, the PNC[55] (https://www.ncbi.nlm.nih.gov/projects/gap/cgi-bin/study.cgi?study_id=phs000607.v1.p1) study, the ADNI[56] (http://adni.loni.usc.edu/data-samples/) study, and ENIGMA2[24] (GWAS of subcortical volumes) and the ENIGMA-CHARGE[34] collaboration (http://enigma.ini.usc.edu/research/). More information about original GWAS design can be found in Zhao et al.[38] and Zhao et al.[39] for UKB, ADNI, HCP, PING, and PNC studies; and in Hibar et al.[24] and Adams et al.[33] for ENIGMA studies. Details about GWAS on validation cohorts (HCP, PING, PNC, ADNI, and ENIGMA) were also provided in Supplementary Note. For discovery, we used the GWAS summary statistics of the UKB study. Then the GWAS results of the other studies were used for validation, see Supplementary Data 23 for a summary of sample size, IDs, names, and modalities of the analyzed neuroimaging traits of each GWAS. To explore genetic overlaps, we also performed TWAS analysis for 16 brain-related complex traits, see Supplementary Data 15 for these data resources.

**Cross-tissue TWAS analysis by UTMOST**. Cross-tissue TWAS analysis was performed for each trait using the UTMOST software (https://github.com/Joker-Jerome/UTMOST). We performed UTMOST analysis using GTEx v6 and v8 reference panels separately. Details about UTMOST model training using GTEx v8 data can be found in Supplementary Note. We first run a single-tissue association test for each GTEx reference panel (44 panels in v6 and 49 panels in v8, respectively) using the above GWAS summary statistics as input. There were 22,694 candidate genes considered in UTMOST. Second, the gene-trait associations in all panels (tissues) were combined by the GBJ test (https://cran.r-project.org/web/packages/GBJ/, R version 3.5.0). We used the pre-trained cross-tissue imputation models and pre-calculated covariance matrices provided by UTMOST. For the 211 neuroimaging traits in the UKB cohort, we also performed a brain tissue-specific version of UTMOST analysis that only combined the brain tissues in GTEx (10 tissues in v6 and 13 tissues in v8, respectively). We applied the Bonferroni correction to account for all candidate genes and traits analyzed in each data set. Specifically, the significance threshold was $5 \times 10^{-2}/22{,}694/211$ in UKB, PING, PNC, and HCP cohorts, $5 \times 10^{-2}/22{,}694/101$ in ADNI cohort, and $5 \times 10^{-2}/22{,}694/16$ in the analysis of 16 other complex traits and clinical outcomes. For each cohort, we obtained a list of significant associations for GTEx v6 and v8 versions, respectively. We reported genes that were either (1) significant in both versions; or (2) significant in one version and at least one of its neighboring (within ±1 MB window) gene was significant in the other version.

**Comparison with previous GWAS findings**. We compared TWAS-significant genes with those identified in the same UKB cohort by MAGMA gene-based association analysis and FUMA functional gene mapping analysis, which can be found in previous GWAS (Supplementary Tables 12 and 15 of Zhao et al.[39] for ROI volumes and Supplementary Tables 14 and 16 of Zhao et al.[40] for DTI parameters, respectively). For each significant gene-trait association, we also explored whether any genetic variant of this gene region (with 1 MB window on both sides) had been linked to this neuroimaging trait by checking the smallest $p$ value in corresponding GWAS. For TWAS-significant genes that were not identified in GWAS, we used NHGRI-EBI GWAS catalog (version 2019-10-14, https://www.ebi.ac.uk/gwas/) to look for their reported associations with brain structure traits and any other traits. We summarized the traits that frequently reported for these genes, such as physical measures (e.g., height, waist-to-hip ratio, heel bone mineral density, body mass index), cognitive functions (such as general cognitive ability, cognitive performance), intelligence, educational attainment, math ability (such as highest math class taken and self-reported math ability), reaction time, neuroticism, neurodegenerative diseases (such as Alzheimer's disease and Parkinson's disease), neuropsychiatric disorders (such as major depressive disorder, schizophrenia, and bipolar disorder), coronary artery disease, and mean corpuscular hemoglobin.

**Cross-tissue analysis conditional on the most significant GWAS signal**. The TWAS gene expression imputation model can be viewed as a weighted sum of multiple genetic variants. If certain variant has a relatively large weight, the imputed gene expression could be driven by a single GWAS signal. In order to look at how many significant TWAS signals could be dominated by a single genetic variant, we rerun TWAS analysis in UKB cohort conditional on the most significant variant used in the UTMOST imputation model (R version 3.5.0). First, for each reference panel, we considered a simple linear model

$Phenotype \sim imputed\ gene\ expression + variant,$

where the variant conditioned on was the most significant variant in previous GWAS of this phenotype in the same UKB cohort. Then, similar to cross-tissue TWAS analysis, single-tissue conditional $p$ values of the imputed gene expression were combined by the GBJ test across the GTEx reference panels (44 panels in GTEx v6 and 49 panels in GTEx v8, respectively).

**Chromatin interaction enrichment analysis**. The chromatin interaction enrichments between significant and non-significant genes were tested using the Wilcoxon rank sum test (R version 3.5.0). For the adult neural Promoter Capture Hi-C, the enrichment of each gene was measured as the number of interactions

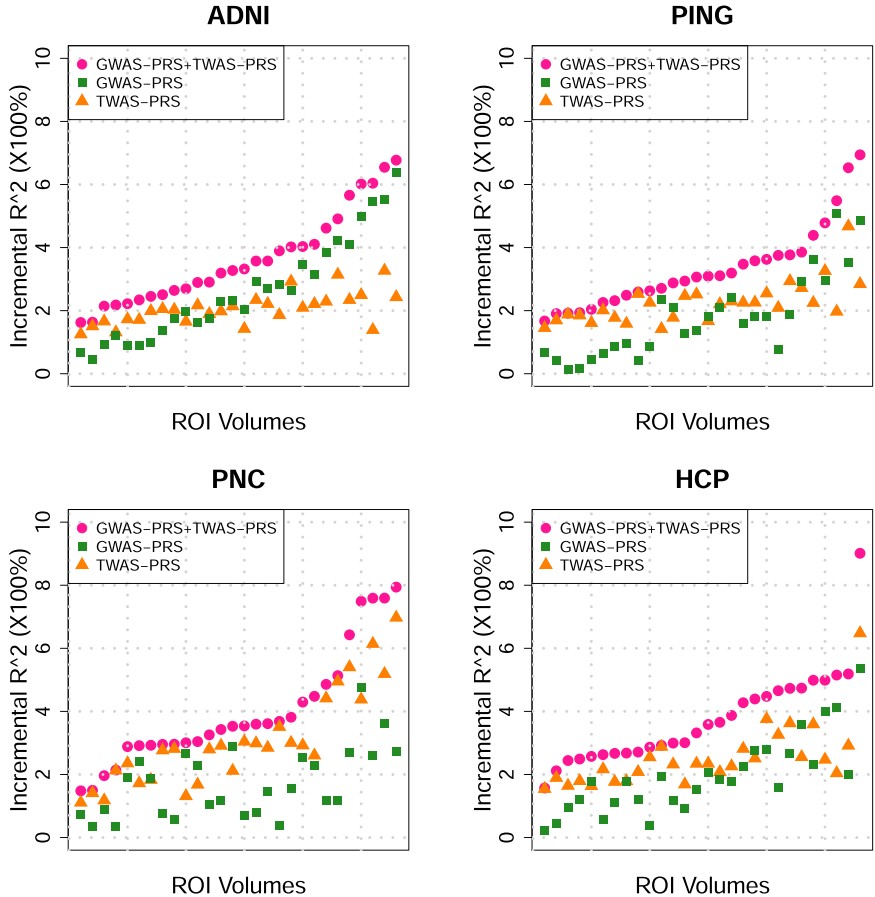

**Fig. 5 Prediction accuracy (incremental $R^2$) of gene-based polygenic risk scores constructed by UKB-derived TWAS summary statistics (TWAS PRS), variant-based PRS constructed by UKB-derived GWAS summary statistics (GWAS PRS), and both of them (GWAS PRS + TWAS PRS) on the four independent data sets ($n$ = 19,629 subjects).** The x axis lists 28 ROI volumes whose TWAS PRS are significant in all the four data sets after the Bonferroni correction and the y axis lists the proportions of phenotypic variation that can be additionally explained by PRS.

overlapping gene with CHiCAGO Enrichment Score >5[96]. The enrichment was tested separately in four cell types, including iPSC-induced excitatory neurons, iPSC-derived hippocampal DG-like neurons, iPSC-induced lower motor neurons, and primary astrocytes. The Wilcoxon rank sum test was separately performed for the significant genes obtained from cross-tissue TWAS analysis, FUMA/MAGMA, and brain tissue-specific TWAS analysis.

**Gene-based TWAS polygenic risk prediction**. Gene-based polygenic profiles were created to assess the out-of-sample prediction power of the UKB TWAS results. In this analysis, we used the individual-level phenotype and genetic data, whose processing steps were detailed in the previous GWAS[39,40]. The FUSION software and database (http://gusevlab.org/projects/fusion/) were used to impute gene expression levels in UKB, ADNI, HCP, PNC, and PING data sets using individual-level genetic data. We performed imputation for 52 different reference panels (Supplementary Data 22). In training data (UKB), we estimated the effect size of each imputed gene expression in a linear regression model, whereas adjusting for the age (at imaging), age-squared, sex, age-sex interaction, age-squared-sex interaction, as well as the top 40 genetic principle components provided by UKB[110] (Data-Field 22009). For ROI volumes, we also included total brain volume (for ROIs other than total brain volume itself) as a covariate. The gene-based TWAS PRS were generated by summarizing across imputed gene expressions, weighed by their effect sizes estimated from the training data. We tried a series of p value thresholds for predictor selection: 1, 0.8, 0.5, 0.4, 0.3, 0.2, 0.1, 0.08, 0.05, 0.02, 0.01, 0.001, $1 \times 10^{-4}$, $1 \times 10^{-5}$, $1 \times 10^{-6}$, $1 \times 10^{-7}$, and $5 \times 10^{-8}$. Thus, 17 polygenic profiles were generated for each neuroimaging trait and we reported the best prediction power that can be achieved by a single profile of them in the single reference panel. The association between polygenic profile and trait was estimated and tested in linear regression model (R version 3.5.0), adjusting for the effects of age and sex. The additional phenotypic variation that can be explained by polygenic profile (i.e., the incremental $R^2$) was used to measure the prediction power. Next, we additionally considered the best variant-based GWAS PRS reported in Zhao et al.[39] and re-evaluated the incremental $R^2$. Specifically, we considered the following four simple linear models

$$Phenotype \sim covariates \ (m1),$$

$$Phenotype \sim TWAS\ PRS + covariates \ (m2),$$

$$Phenotype \sim GWAS\ PRS + covariates \ (m3), \text{ and}$$

$$Phenotype \sim TWAS\ PRS + GWAS\ PRS + covariates \ (m4).$$

We estimated the incremental $R^2$ of TWAS PRS conditioning on GWAS PRS using models m4 and m3, the incremental $R^2$ of GWAS PRS conditioning on TWAS PRS using models m4 and m2, and calculated the additional phenotypic variation that can be jointly explained by GWAS and TWAS PRS using models m4 and m1. More details about constructing and evaluating gene-based PRS can be found in Supplementary Note.

**Reporting summary**. Further information on research design is available in the Nature Research Reporting Summary linked to this article.

## Data availability

The data used in this work were obtained from publicly available data sets: the UK Biobank (UKB) study, the Human Connectome Project (HCP) study, the Pediatric Imaging, Neurocognition, and Genetics (PING) study, the Philadelphia Neurodevelopmental Cohort (PNC) study, the Alzheimer's Disease Neuroimaging Initiative (ADNI) study, and ENIGMA2 & the ENIGMA-CHARGE collaboration. For the first five data sets, the raw MRI, covariates, and SNP data are available from each data resource: UK Biobank, http://www.ukbiobank.ac.uk/resources/;PING, http://pingstudy.ucsd.edu/resources/genomics-core.html/; PNC, https://www.ncbi.nlm.nih.gov/projects/gap/cgi-bin/study.cgi?study_id=phs000607.v1.p1/; ADNI, http://adni.loni.usc.edu/data-samples/; and HCP, https://www.humanconnectome.org/. The GWAS summary statistics can be obtained at https://github.com/BIG-S2/GWAS and http://enigma.ini.usc.edu/research/. In addition, we used other 16 sets of publicly available GWAS summary statistics shared by several GWAS databases. These data resources are summarized in Supplementary Data 15. The FUSION database used in this study is available at http://gusevlab.org/projects/fusion/.

## Code availability

We made use of publicly available software and tools, especially the UTMOST ([https://github.com/Joker-Jerome/UTMOST](https://github.com/Joker-Jerome/UTMOST)) and the FUSION ([http://gusevlab.org/projects/fusion/](http://gusevlab.org/projects/fusion/)). The analysis code is freely available at [https://doi.org/10.5281/zenodo.4649360](https://doi.org/10.5281/zenodo.4649360)[111].

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

## Acknowledgements

This research was partially supported by U.S. NIH grants MH086633 (HT.Z.), HD079124 (Y.L.), HL129132 (Y.L.), and MH116527 (TF.L.). We thank Quan Wang, Bingshan Li, and Jia Wen for helpful conversations. We thank the individuals represented in the UK Biobank, ADNI, HCP, PING, PNC, ENIGMA2, and ENIGMA-CHARGE data sets for their participation and the research teams for their work in collecting, processing, and disseminating these data sets for analysis. This research has been conducted using the UK Biobank resource (application number 22783), subject to a data transfer agreement. We gratefully acknowledge all the studies and databases that made GWAS summary data available. The data resources had obtained informed consent from all participants and had obtained approval from their research ethics committees or institutional review boards. The UKB study had obtained ethics approval from the North West Multicentre Research Ethics Committee (approval number: 11/NW/0382). ADNI study was approved by all the institutional ethical review boards of all participating

centers. The institutional review boards of the University of Pennsylvania and the Children's Hospital of Philadelphia approved all study procedures in the PNC study. The human research protection programs and institutional review boards at the nine institutions participating in the PING project approved all experimental and consenting procedures. All experimental procedures in the HCP study were approved by the institutional review boards at Washington University (approval number: 201204036). Part of data collection and sharing for this project was funded by the Alzheimer's Disease Neuroimaging initiative (ADNI) (National Institutes of Health Grant U01 AG024904) and DOD ADNI (Department of Defense award number W81XWH-12-2-0012). ADNI is funded by the National Institute on Aging, the National Institute of Biomedical Imaging and Bioengineering and through generous contributions from the following: Alzheimer's Association; Alzheimer's Drug Discovery Foundation; Araclon Biotech; BioClinica, Inc.; Biogen Idec Inc.; Bristol-Myers Squibb Company; Eisai Inc.; Elan Pharmaceuticals, Inc.; Eli Lilly and Company; EuroImmun; F. Hoffmann-La Roche Ltd and its affiliated company Genentech, Inc.; Fujirebio; GE Healthcare; IXICO Ltd; Janssen Alzheimer Immunotherapy Research & Development, LLC; Johnson & Johnson Pharmaceutical Research & Development LLC; Medpace, Inc.; Merck & Co., Inc.; Meso Scale Diagnostics, LLC; NeuroRx Research; Neurotrack Technologies; Novartis Pharmaceuticals Corporation; Pfizer Inc.; Piramal Imaging; Servier; Synarc Inc.; and Takeda Pharmaceutical Company. The Canadian Institutes of Health Research is providing funds to support ADNI clinical sites in Canada. Private sector contributions are facilitated by the Foundation for the National Institutes of Health (www.fnih.org). The grantee organization is the Northern California Institute for Research and Education, and the study is coordinated by the Alzheimer's Disease Cooperative Study at the University of California, San Diego. ADNI data are disseminated by the Laboratory for Neuro Imaging at the University of Southern California. Part of the data collection and sharing for this project was funded by the Pediatric Imaging, Neurocognition and Genetics Study (PING) (U.S. National Institutes of Health Grant RC2DA029475). PING is funded by the National Institute on Drug Abuse and the Eunice Kennedy Shriver National Institute of Child Health & Human Development. PING data are disseminated by the PING Coordinating Center at the Center for Human Development, University of California, San Diego. Support for the collection of the PNC data sets was provided by grant RC2MH089983 awarded to Raquel Gur and RC2MH089924 awarded to Hakon Hakonarson. All PNC subjects were recruited through the Center for Applied Genomics at The Children's Hospital in Philadelphia. HCP data were provided by the Human Connectome Project, WU-Minn Consortium (Principal Investigators: David Van Essen and Kamil Ugurbil; 1U54MH091657) funded by the 16 NIH Institutes and Centers that support the NIH Blueprint for Neuroscience Research; and by the McDonnell Center for Systems Neuroscience at Washington University.

## Author contributions

B.Z., Y.S., Y.L., and HT.Z. designed the study. B.Z., Y.S., Y.Y., Z.Y., HY.Z., P.S. TF.L., X.W., TY.L., and Z.Z performed the experiments and analyzed the data. B.Z., Y.S., Y.L., and HT.Z. wrote the manuscript with feedback from all authors.

## Competing interests

The authors declare no competing interests.
