## [Peer Review File · Nature Communications]

Reviewers' Comments:

Reviewer #1:

Remarks to the Author:

In this submission, Zhao et al leverage imputed transcriptome levels to survey evidence of genetic regulation on structural variation of human brain, and consequences on brain function. They use a state-of-the-art TWAS method leveraging cross-tissue information (UTMOST) on 211 imaging traits from the UK Biobank cohort, validate the results in 5 independent studies, and complement with gene-based polygenic risk scores. To assay relationships to other brain-related traits or function, additional TWAS analysis on 11 separate neuropsychiatric traits is conducted. The results are appropriately discussed in the context of current literature.

I think this paper analyzes an important problem, the technical approach is sound and the exposition is adequate in most of the text. However, I have a few concerns.

Major comments:

1) The analysis leans heavily on gene-based association methods, which are notoriously vulnerable in scenarios with high linkage disequilibrium (LD) and pleiotropic effects. I.e. The associations might be spurious due to expression and traits being affected by distinct variants that are in LD. This becomes even more complex in the gene-based PRS analysis. I don't think that merely acknowledging this issue in the discussion is enough.

To make the results more convincing, I strongly suggest complementing the analysis with a colocalization measure such as ENLOC (Wen et al, Plos Genetics 2017) or eCAVIAR (Hormozdiari et al, AJHG 2016). For example, add verification of which TWAS associations have a colocalized signal in any tissue (or any brain tissue).

I think the gene-based PRS section might benefit too from colocalization analysis. The simplest possibility could be a GWAS-to-GWAS colocalization analysis, and a more ambitious approach could use MOLOC (Giambartolomei et al, Biostatistics 2018) to test (training/UKB GWAS; expression; testing/non UKB GWAS) tuples.

2) The UTMOST-based analysis uses GTEx v6 models. This data release is quite dated, and since then, the GTEx consortium significantly changed the genotyping, transcript quantification and RNA-seq alignment methods used. I suggest updating the UTMOST-based analysis at least to GTEx v7; especially since the gene-based PRS analysis uses GTEx v7 ([page 10, line 28]; methods [page 26, line 16]). This will make the analysis more consistent and convincing.

3) I couldn't find an explicit description of the 211 UK Biobank neuroimaging traits (such as names, modality, abbreviations/IDS used in the rest of the manuscript). Please clarify if there is one and I missed it, or otherwise add a table with this information to make the submission self contained.

4) [page 10, line 8]: about the sentence "We next performed some functional lookups": It is unclear to me what was meant. E.g. was FUMA used in some specific way? Please clarify.

5) [page 11, line 13]: It is unclear to me what "put both GWAS and TWAS PRS in one model" means. The rest of the paragraph is also unclear to me.

Was a OLS linear model with 2 explanatory variables (gene-based PRS and GWAS-based PRS) used to predict phenotype?

i.e. Is this text explained by [page 27, line 7]?

Please clarify.

Minor comments:

1) In the brain tissue-specific section of results, please mention the number of brain tissues. (10 for GTEx v6?)

2) In the second paragraph of "Brain tissue-specific section" [page 7, line 23], the language seems to switch to "tissue-specific".

If this "tissue-specific" wording refers to "brain tissue-specific" analysis, I suggest keeping the "brain" word for clarity.

If this "tissue-specific" wording refers to something else, then it is unclear to me what is being referred to.

3) In the section "Compared to brain tissue-specific TWAS analysis" section, I would make the scientific questions more explicit. E.g. state "comparing power to detect association between brain tissues and all tissues"; "which genes can be identified by all tissues vs brain tissues", etc.

4) [page 8, line 12]: I believe the text should read "p-value of the the most significant variant was smaller than 1×10^{-6} ".

5) Methods [page 23, line 30]: I suggest moving the appropriate references on [page 24, line 12] (24, 33, 38, 39) adjacent to each cohort being mentioned.

Reviewer #2:

Remarks to the Author:

The current manuscript used transcriptome-wide association study (TWAS) methods to link gene expression to existing genome-wide associations of 211 neuroimaging traits. With this method they were able to find 204 associated genes (based on gene expression) of which 86 genes were novel (not previously linked to brain structure) using data from the UK Biobank cohort. They used a discovery-validation strategy using 5 other datasets. They were able to validate 10 novel genes with this strategy. They also investigated polygenic risk score analysis (PRS) based on this TWAS approach (TWAS PRS) and compared if prediction abilities of traditional (genetic association based) PRS analysis improved if TWAS PRS were added, which was indeed what they found. In the manuscript they describe the (novel) genes and the known associations to other phenotypes of these genes, as well as comparisons between tissue-specific TWAS approaches and cross-tissue TWAS approaches (the latter is what they used as the main analysis in the current manuscript). They conclude that TWAS can improve our interpretation of genome-wide association studies of neuroimaging traits, find novel genes and in particular if cross tissue approaches are being used.

The manuscript reports novel and interesting results that will be interesting to scientists in the field of imaging genetics. I only have a few points that need clarification or a broader perspective, in my opinion.

1. It is unclear from the description how the TWAS PRS is made (now it is only described that the FUSION program does this by summarizing across imputed gene expressions, weighted by their effect size estimated from the training data). A few things are unclear, please specify which imputed gene expression (based on which data?), which values are being aggregated to get to the PRS score, and whether this is thresholded?, and which effect sizes from the training set (based on which analysis?)? And in particular how here the tissue specificity vs the cross-tissue analysis holds?

2. More information is needed for the (smaller) validation cohorts that are being used. For details on the validation samples (such as imputation, association, ethnicity, quality control) they are now referring to the references of these papers, but they only include the larger samples (like ENIGMA) and not the smaller samples (like HCP and PNC). Did they run the QC and genome-wide association analysis themselves for these samples? (if so this needs to be stated, particularly because the genetic analysis of these samples is not trivial, for example both samples have ethnicity challenges). Even if they did not do this themselves the reader needs to be able to know some more details on these

samples that should be part of this manuscript (and does not rely on readers looking for this data in other references).

3. The results mention how the TWAS associated genes were previously related to "brain-related traits" and "neuropsychiatric traits", can the authors put this in perspective to "other traits"? (are these genes just associated to many traits or are they specific for brain/neuropsychiatric traits?)

4. The authors conclude that the cross tissue analysis is more powerful than the tissue specific TWAS. Is this fact that the cross tissue analysis is based on the largest possible number a factor in this? Can the authors comment on that?

5. Is the gene-wide approach used in TWAS appropriate in all cases here? (Based on the findings that 418 of the 614 associations are dominated by the strongest GWAS signal)

6. The enrichment analysis in promotor-related chromatin interactions of four types of brain regions is found to be enriched in all datasets. Is it possible to compare this in some way to non-brain specific sets? (again to compare brain specificity vs more general processes)

7. They include 11 neuropsychiatric traits, but it is not specified how these 11 traits were selected? (and they are not an obvious selection)

8. In the discussion they mention that "brain tissue reference panels do not have large sample sizes" and how this is a limitation. However, this is also the case for many of the other reference samples right? And this step (from reference sample to expression imputation) is not covered at all in the current paper. For example what happens to genetic variants with very low frequency if the reference panel is very small? (For example if the expression is based on samples smaller than 100 individuals, and the genetic variant is only carried by one or two of the individuals?)

Transcriptome-wide association analysis of 211 neuroimaging traits identifies new genes for brain structures and yields insights into gene-level pleiotropy with complex neuropsychiatric traits

Responses to the Reviewer 1:

Thank you for your careful review and constructive suggestions! Here we provide our point-to-point responses. We have made changes in the manuscript accordingly, with changes highlighted. For your convenience, we first restate your comments in italic and then provide our responses.

In this submission, Zhao et al leverage imputed transcriptome levels to survey evidence of genetic regulation on structural variation of human brain, and consequences on brain function. They use a state-of-the-art TWAS method leveraging cross-tissue information (UTMOST) on 211 imaging traits from the UK Biobank cohort, validate the results in 5 independent studies, and complement with gene-based polygenic risk scores. To assay relationships to other brain-related traits or function, additional TWAS analysis on 11 separate neuropsychiatric traits is conducted. The results are appropriately discussed in the context of current literature. I think this paper analyzes an important problem, the technical approach is sound and the exposition is adequate in most of the text. However, I have a few concerns.

Response: Many thanks for your supportive comments! We cannot agree with you more on the importance of our research question and have strived to do our best to revise this paper.

Major comments:

(1) The analysis leans heavily on gene-based association methods, which are notoriously vulnerable in scenarios with high linkage disequilibrium (LD) and pleiotropic effects. I.e. The associations might be spurious due to expression and traits being affected by distinct variants that are in LD. This becomes even more complex in the gene-based PRS analysis. I don't think that merely acknowledging this issue in the discussion is enough.

To make the results more convincing, I strongly suggest complementing the analysis with a colocalization measure such as ENLOC (Wen et al, Plos Genetics 2017) or eCAVIAR (Hormozdiari et al, AJHG 2016). For example, add verification of which TWAS associations have a colocalized signal in any tissue (or any brain tissue).

I think the gene-based PRS section might benefit too from colocalization analysis. The simplest possibility could be a GWAS-to-GWAS colocalization analysis, and a more

ambitious approach could use MOLOC (Giambartolomei et al, Biostatistics 2018) to test (training/UKB GWAS; expression; testing/non UKB GWAS) tuples.

Response:

Thank you very much for your in-depth thoughts and comments! Following your suggestions, we have performed colocalization analyses to 1) complement our TWAS analysis with a colocalization measure, and to 2) explore whether we can improve the performance of gene-based PRS.

Part I: Complementing TWAS Analysis with a Colocalization Measure

In our understanding, you would like us to complement our analyses with colocalization measures to highlight the TWAS signals that can also be detected by colocalization analysis, which are presumably less likely to be spurious results due to variants in LD. Following your suggestion, we have performed colocalization analysis for each of the 918 gene-trait pairs (278 genes) identified in our cross-tissue TWAS analysis. Specifically, we used fastENLOC (<https://github.com/xqwen/fastenloc>) for colocalization analysis, which is a faster version of the ENLOC method suggested in your comment. We chose ENLOC/fastENLOC over eCAVIAR because the model implemented in ENLOC is a more general framework, in which the eCAVIAR method can be viewed as a special case with a more restricted assumption. In particular, we applied fastENLOC to our UK Biobank GWAS summary statistics with eQTL analysis results from each of the 49 GTEx v8 tissues provided by fastENLOC. For the GWAS summary statistics input, LD blocks were annotated based on the European-based LD files, which were also provided by fastENLOC.

We then obtained the regional colocalization probability (RCP) of each of the 49 GTEx tissues for each of our 918 reported gene-trait TWAS signals. Among the 918 gene-trait pairs reported by our cross-tissue TWAS analysis, 103 (11.22%) of them had RCP > 0.5 in at least one GTEx tissue, which were related to 45 of the 278 (16.2%) TWAS-significant genes. We found that the maximum RCP (across 49 tissues) located in one of the 13 brain tissues for 35 of the 103 colocalization signals. Across all signals with RCP > 0.5 in at least one tissue, the mean RCP is 0.73. We found a few interesting signals that were both detected by TWAS and colocalization analysis. For example, the gene *SLC16A8* is known to be a risk factor of glioblastoma/glioma and we found that this gene was significant in our TWAS analysis, and it also had very high RCP (0.92) in the colocalization analysis. Overall, the overlapped signals detected by both methods give us

more confident in those signals (103 associations, 45 genes). We have reported these results in Supplementary Table 14. It is also worth to mention that the UTMOST TWAS analysis uses a cross-tissue testing approach, whereas the fastENLOC colocalization analysis is a single-tissue analysis. It is therefore expected that the results of UTMOST and fastENLOC may not perfectly overlap. We have updated our manuscript as follows:

“Next, we applied fastENLOC to perform colocalization analysis for the 278 cross-tissue TWAS-significant genes (Methods). We found that 45 of the 278 (16.2%) genes (involving 103 of 918 gene-trait associations) had regional colocalization probability (RCP) > 0.5 in at least one tissue type and seven genes (involving 17 gene-trait associations) had RCP > 0.9 (Supplementary Table 14). Among them, there are known risk genes. For example, SLC16A8 is a known risk gene of glioma/glioblastomas. In our cross-tissue TWAS analysis, SLC16A8 was significantly associated with multiple white matter microstructures, and fastENLOC colocalization analysis also found that SLC16A8 had a high colocalization probability (0.919) with expression quantitative trait loci (eQTL) signals in GTEx v8 nerve tibial tissue type.”

(Page 10, Lines 20-29)

Part II: Gene-Based PRS with Colocalization

We first would like to clarify a few comments. Regarding your suggestion on ‘the PRS section might benefit too from colocalization analysis’, this benefit may reflect through a higher prediction accuracy (% variation in the trait additionally explained by the constructed gene-based PRS) by utilizing colocalization measures in constructing the gene-based PRS. With regard to the simpler suggestion ‘GWAS-to-GWAS colocalization analysis’, we understand it as a simpler version of the ‘more ambitious’ suggestion ‘to test (training/UKB GWAS; expression; testing/non UKB GWAS) tuples’ using MOLOC. Therefore, we think that the ‘GWAS-to-GWAS colocalization analysis’ means a colocalization analysis between training (UKB) GWAS and the testing (non-UKB) GWAS results, which can inform us on the agreement between two GWAS studies. As for the more ambitious MOLOC (Giambartolomei et al, Biostatistics 2018) analysis on the tuples, a third group of information, the expression, is also considered. As we understand, ‘expression’ here refers to the eQTL analysis results from GTEx, which is used in our TWAS analysis; and the tuple is the three portions of information used in our TWAS analysis.

We therefore implemented the more ambitious approach using MOLOC (Giambartolomei et al, Biostatistics 2018). We input the UKB GWAS summary statistics and meta-analyzed non-UKB GWAS summary statistics, and then performed MOLOC using each of the 49 GTEx v8 tissues separately for all of our 211 neuroimaging traits. For each gene, MOLOC outputs the posterior probability (PP) of having a colocalized signal shared among the three datasets (UKB GWAS, non-UKB GWAS, and GTEx eQTL). We used this PP to further weight the genes when constructing the gene-based PRS. Specifically, we let MOLOC-weighted gene-based PRS to be $\sum_i w_i \cdot \hat{\beta}_i \cdot \text{gene}_i$, where w_i is the maximum of PP across different reference panels for the i th gene, $\hat{\beta}_i$ is the estimated gene effect size of the i th gene from the training data (UKB GWAS), and gene_i is the imputed gene expression of the i th gene in the testing data (non-UKB GWAS). The only difference between MOLOC-weighted PRS and our original gene-level PRS is that our original gene-level PRS did not use w_i . For genes not present in the MOLOC results, we set w_i to be $0.5 \cdot \text{minimum PP across all the genes}$. Intuitively, w_i uses the colocalization information to weight these genes and prioritizes the genes with high PP.

The performance of MOLOC-weighted PRS for our neuroimaging traits is shown in the following figure (right panel, mean R-squared = 2.27%), which is similar to the performance of our original gene-level PRS (left panel, mean R-squared = 2.34%). Overall, our results suggest that using the colocalization results in our gene-based PRS did not result in higher prediction performance (Wilcoxon rank test p-value = 0.88).

(2) The UTMOST-based analysis uses GTEx v6 models. This data release is quite dated, and since then, the GTEx consortium significantly changed the genotyping, transcript quantification and RNA-seq alignment methods used. I suggest updating the UTMOST-based analysis at least to GTEx v7; especially since the gene-based PRS analysis uses GTEx v7 ([page 10, line 28]; methods [page 26, line 16]). This will make the analysis more consistent and convincing.

Response: Many thanks for pointing this out! To address this issue, we have worked with the authors of UTMOST (Professor Hongyu Zhao's Lab from Yale) to retrain all the UTMOST models using the GTEx v8 database. The UTMOST-GTEx-v6 models identified 204 genes (614 gene-trait associations). We found that the new UTMOST-GTEx-v8 models identified 348 genes (1177 gene-trait associations). Of the 204 GTEx-v6 significant genes, 45 genes (22.1%) were also significant in the GTEx-v8 version, and 152 genes (74.5%) were overlapped (with 1MB window on both sizes) with at least one GTEx-v8 significant gene. The difference between results from the two versions could be partially explained by the differences of the genotyping and RNA-seq techniques in the GTEx data.

To make our results robust to the GTEx data versions, we reported genes that were either 1) significant in both of the two versions; or 2) significant in one version and were overlapped (within 1MB window on both sizes) with at least one significant gene in the other version. This procedure was used throughout our updated manuscript and we have revised the whole paper accordingly. Overall, we discovered and validated more significant associations using both GTEx v6 and v8 data. For example, in the cross-trait TWAS analysis for UKB discovery dataset (our main analysis), this time we identified 918 significant gene-trait associations between 278 genes and 152 neuroimaging traits, which were significantly larger than the ones only using GTEx v6 data (614 associations for 204 genes). And we can validate 29 genes in the updated manuscript, which was also much larger than our previous results (18 genes).

“The original version of UTMOST models was trained using GTEx v6 as the reference. In this study, we retrained the UTMOST models using the recently released GTEx v8 data and performed our analysis using both versions. In the rest of this paper, we reported genes that were either 1) significant in both versions; or 2) significant in one version and were overlapped (within $\pm 1\text{MB}$ window) with at least one significant gene in the other version.”

(Page 5, Lines 10-15)

(3) I couldn't find an explicit description of the 211 UK Biobank neuroimaging traits (such as names, modality, abbreviations/IDS used in the rest of the manuscript). Please clarify if there is one and I missed it, or otherwise add a table with this information to make the submission self-contained.

Response: We apologize for the confusion. We put the IDS in Supplementary Table 23 in the previous version, but we did not make it very clear. We have highlighted this in the Method section in this revision. Moreover, following your suggestion, we have also added the original names and modality of IDS in Supplementary Table 23.

“...see Supplementary Table 23 for a summary of sample sizes, IDs, names, and modalities of the analyzed neuroimaging traits of each GWAS.”
(Page 25, Lines 28-29)

(4) [page 10, line 8]: about the sentence "We next performed some functional lookups": It is unclear to me what was meant. E.g. was FUMA used in some specific way? Please clarify.

Response: We are sorry for this confusing wording. We meant to say that we used some additional genomics data from literature to support and highlight some of our interesting findings. Moreover, FUMA was not used here. To avoid such confusion, we have removed “functional lookups” and changed it to

“We next performed some additional analysis for the 19 validated UKB TWAS novel genes. First, ...”
(Page 11, Lines 16-17)

(5) [page 11, line 13]: It is unclear to me what "put both GWAS and TWAS PRS in one model" means. The rest of the paragraph is also unclear to me. Was a OLS linear model with 2 explanatory variables (gene-based PRS and GWAS-based PRS) used to predict phenotype? i.e. Is this text explained by [page 27, line 7]? Please clarify.

Response: We are sorry for this confusing wording. We indeed used a simple OLS linear model with both gene-based PRS and GWAS-based PRS to predict the phenotype. We

explained this by using models (m1) and (m4) around Page 27, line 7 (in the old version, now Page 28, Lines 29-32). We have provided more details in that paragraph:

“...we next include both GWAS and TWAS PRS as predictors in one linear model to predict the above 28 TWAS-significant ROI volumes (Method).”

(Page 12, Lines 22-23)

In addition, we have added a new section in Supplementary Note (Section 3.1) to describe details on gene-based PRS analysis:

“More details about constructing and evaluating gene-based PRS can be found in Section 3.1 of Supplementary Note.”

(Page 29, Lines 4-6)

Minor comments:

(1) In the brain tissue-specific section of results, please mention the number of brain tissues. (10 for GTEx v6?)

Response: Thank you for your suggestion! There are 10 brain tissues in GTEx v6 and 13 brain tissues in GTEx v8. We have included this information in the tissue-specific section:

“As a comparison, we performed a brain tissue-specific version of UTMOST TWAS that only combined brain tissues (10 brain tissues in GTEx v6 or 13 brain tissues in GTEx v8, Method).”

(Page 7, Lines 28-30)

(2) In the second paragraph of "Brain tissue-specific section" [page 7, line 23], the language seems to switch to "tissue-specific". If this "tissue-specific" wording refers to "brain tissue-specific" analysis, I suggest keeping the "brain" word for clarity. If this "tissue-specific" wording refers to something else, then it is unclear to me what is being referred to.

Response: Thanks a lot! Following your suggestion, we have replaced “tissue-specific” by “brain tissue-specific” in the whole manuscript.

(3) In the section "Compared to brain tissue-specific TWAS analysis" section, I would make the scientific questions more explicit. E.g. state "comparing power to detect association between brain tissues and all tissues"; "which genes can be identified by all tissues vs brain tissues", etc.

Response: Thank you for your suggestion! We have changed the section title to “Comparing power to detect association between brain tissues and all tissues” in the manuscript. (Page 7, Line 27)

(4) [page 8, line 12]: I believe the text should read "p-value of the most significant variant was smaller than 1×10^{-6} ".

Response: Thank you for pointing this out! We just highlighted the existence of genes that have *large* GWAS p-value, which is larger than the stringent threshold, and thus are difficult to be detected (i.e., to become significant) in GWAS. Thus, we checked the *smallest* p-value of genetic variants within the gene (i.e., the most significant variants), and stated that these variants were still not significant enough to be detected in GWAS (i.e., p-values were still *too large*). We have changed our sentence to make this clearer:

“The GWAS p-value of the most significant variant (i.e., the variant with the smallest p-value) was greater than 1×10^{-6} for associations of 19 genes”

(Page 8, Lines 28-29)

(5) Methods [page 23, line 30]: I suggest moving the appropriate references on [page 24, line 12] (24, 33, 38, 39) adjacent to each cohort being mentioned.

Response: Thank you for your careful review! As you suggested, we have moved these references to the place where they were introduced (Page 25, Lines 12-24). In addition, we have provided more details about these GWAS in Supplementary Note:

“Details about GWAS on validation cohorts (HCP, PING, PNC, ADNI, and ENIGMA) were also provided in Section 3.2 of Supplementary Note.”

(Page 25, Lines 24-26)

Response to the Reviewer 2:

Thank you for your careful review and constructive suggestions! Here we provide our point-to-point responses. We have made changes in the manuscript accordingly, with changes highlighted. For your convenience, we first restate your comments in italic and then provide our responses.

The current manuscript used transcriptome-wide association study (TWAS) methods to link gene expression to existing genome-wide associations of 211 neuroimaging traits. With this method they were able to find 204 associated genes (based on gene expression) of which 86 genes were novel (not previously linked to brain structure) using data from the UK Biobank cohort. They used a discovery-validation strategy using 5 other datasets. They were able to validate 10 novel genes with this strategy. They also investigated polygenic risk score analysis (PRS) based on this TWAS approach (TWAS PRS) and compared if prediction abilities of traditional (genetic association based) PRS analysis improved if TWAS PRS were added, which was indeed what they found. In the manuscript they describe the (novel) genes and the known associations to other phenotypes of these genes, as well as comparisons between tissue-specific TWAS approaches and cross-tissue TWAS approaches (the latter is what they used as the main analysis in the current manuscript). They conclude that TWAS can improve our interpretation of genome-wide association studies of neuroimaging traits, find novel genes and in particular if cross tissue approaches are being used. The manuscript reports novel and interesting results that will be interesting to scientists in the field of imaging genetics. I only have a few points that need clarification or a broader perspective, in my opinion.

Response: Many thanks for your supportive comments! Based on your comments and suggestions, we have tried our best to revise this paper as detailed below.

(1) It is unclear from the description how the TWAS PRS is made (now it is only described that the FUSION program does this by summarizing across imputed gene expressions, weighted by their effect size estimated from the training data). A few things are unclear, please specify which imputed gene expression (based on which data?), which values are being aggregated to get to the PRS score, and whether this is thresholded? and which effect sizes from the training set (based on which analysis)? And in particular how here the tissue specificity vs the cross-tissue analysis holds?

Response: Thank you for your suggestions! We have added a new section in Supplementary Note (Section 3.1) to provide more information about gene-based PRS. As you suggested, we detailed the steps to 1) obtain imputed gene expression; 2) estimate the effect size of imputed gene expression; 3) construct the gene-based PRS; and 4) evaluate the gene-based PRS:

“More details about constructing and evaluating gene-based PRS can be found in Section 3.1 of Supplementary Note.”

(Page 29, Lines 4-6)

Here we briefly summarize the key points in each step:

1. Obtain imputed gene expression:

We used FUSION (<http://gusevlab.org/projects/fusion/>) and plink to impute gene expression levels for each of the 52 non-TCGA reference panels (13 GETx v7 brain tissues, 35 GTEx v7 other tissues, 1 non-GETx brain tissue, and 3 non-GETx other tissues).

2. Estimate the effect size of imputed gene expression:

We used the UKB dataset as our training data to estimate the effect size of each imputed gene expression for each of the 52 reference panels in linear models (with covariates adjusted).

3. Construct the gene-based PRS:

With effect sizes estimated from the UKB dataset, we generated the gene-based TWAS PRS in ADNI, HCP, PNC, and PING datasets by summarizing across imputed gene expressions, weighted by their effect sizes. Multiple p-value thresholds were tried for gene selection.

4. Evaluate the gene-based PRS:

We evaluated the prediction performance by the phenotypic variation that can be additionally explained by gene-based PRS. We reported the best performance across different p-value thresholds and 52 different reference panels.

In our gene-based PRS section, we constructed and evaluated PRS separately for each tissue/panel. We reported the best performance that can be achieved by one single tissue/panel. We also compared the performance of these tissues/panels and found a significant positive relationship between the panel sample size and prediction accuracy (Page 13, Lines 5-14). Cross-tissue gene-based PRS (i.e., constructing gene-based PRS simultaneously using all tissues/panels) might be an interesting future topic to explore.

(2) More information is needed for the (smaller) validation cohorts that are being used. For details on the validation samples (such as imputation, association, ethnicity, quality control) they are now referring to the references of these papers, but they only include the larger samples (like ENIGMA) and not the smaller samples (like HCP and PNC). Did they run the QC and genome-wide association analysis themselves for these samples? (if so this needs to be stated, particularly because the genetic analysis of these samples is not trivial, for example both samples have ethnicity challenges). Even if they did not do this themselves the reader needs to be able to know some more details on these samples that should be part of this manuscript (and does not rely on readers looking for this data in other references).

Response: Many thanks for your insightful comments! We have added a new section in Supplementary Note (Section 3.2) to provide information about validation samples (ADNI, HCP, PNC, PING, and ENIGMA):

“Details about GWAS on validation cohorts (HCP, PING, PNC, ADNI, and ENIGMA) were also provided in Section 3.2 of Supplementary Note.”

(Page 25, Lines 24-26)

Specifically, in HCP, PING, PNC, and ADNI, only subjects of European ancestry were considered in GWAS and standard genetic data quality control was performed.

(3) The results mention how the TWAS associated genes were previously related to “brain-related traits” and “neuropsychiatric traits”, can the authors put this in perspective to “other traits”? (are these genes just associated to many traits or are they specific for brain/neuropsychiatric traits?)

Response: Thanks for the in-depth thoughts and suggestions! Overall, we found that our TWAS-significant genes had been linked to a wide range of complex traits in different domains, including both brain-related traits and non-brain traits (for example, as we showed in Figure 2). As you suggested, we have checked and found that some genes were associated with many traits, while other genes were particularly related to brain-related traits. Thus, both situations you mentioned exist in these genes. For example:

“Moreover, we found that DCC, MIR1-1HG, DPP4, and RECQL4 were specifically associated with brain-related traits and disorders, whereas other genes (such as NUP210L, DLG2, AC090666.1, KCNH7, and JPH3) were also widely associated with non-brain traits, including triglycerides, mean platelet volume, and coronary artery disease.”

(Page 7, Lines 19-23)

(4) The authors conclude that the cross-tissue analysis is more powerful than the tissue specific TWAS. Is this fact that the cross-tissue analysis is based on the largest possible number a factor in this? Can the authors comment on that?

Response: Thanks a lot for your insightful comments! In our understanding, the “number” you mentioned means “sample size”. We believe that tissue/panel sample size is an important factor for the power of TWAS analysis, and cross-tissue analysis combines all tissues and thus always includes the tissues with the largest sample sizes. We have commented on this in the discussion section:

“The better performance of cross-tissue analysis may be partially explained by the fact that cross-tissue analysis always includes tissues with large sample sizes, and reference panel sample size is an important factor in TWAS model fitting and testing power.”

(Page 14, Lines 8-11)

(5) Is the gene-wide approach used in TWAS appropriate in all cases here? (Based on the findings that 418 of the 614 associations are dominated by the strongest GWAS signal)

Response: Thanks a lot for your in-depth comments! Our results may suggest that TWAS is not always appropriate, since sometime the TWAS signals are dominated by the strongest GWAS signal. In these situations, TWAS does not really “aggregate” the signals from multiple variants. We have commented this in the manuscript:

“These results also suggest that TWAS analysis may not always be appropriate for all genes. It is more valuable and can reveal more additional insights when the TWAS signals are not driven by one single genetic variant.”

(Page 9, Lines 26-29)

(6) The enrichment analysis in promotor-related chromatin interactions of four types of brain regions is found to be enriched in all datasets. Is it possible to compare this in some way to non-brain specific sets? (again to compare brain specificity vs more general processes)

Response: Thank you for your suggestions! In our understanding, you suggested us to compare the difference between brain-tissue specific and cross-tissue approaches in enrichment analysis. The enrichment analysis has been separately performed on both the TWAS results from our cross-tissue analysis and brain-tissue specific analysis. We found that cross-tissue analysis leads to more significant enrichment results (p-value range =

[2.3×10^{-2} , 6.18×10^{-2}]). By contrast, we found that the 134 brain-tissue specific genes showed no significant enrichment in all of the four type of brain cells (p-value range = [0.14, 0.68]). We have made some comments on this in the manuscript:

“In contrast, brain tissue-specific TWAS-significant genes did not show any significant enrichment (p-value range = [0.14, 0.68]), indicating the value of cross-tissue TWAS over brain tissue-specific TWAS.”

(Page 10, Lines 15-18)

(7) They include 11 neuropsychiatric traits, but it is not specified how these 11 traits were selected? (and they are not an obvious selection)

Response: Thank you for pointing this out! We selected some recent GWAS of brain-related complex traits with large sample size. We have included five more traits (Alzheimer's Disease, bipolar disorder, T2D, stroke, high blood pressure) in the revised manuscript and added the following explanations:

“...we performed cross-tissue TWAS analysis for 16 other brain-related complex traits with large GWAS sample size, including neuropsychiatric traits, cognition, and cardiovascular risk factors”

(Page 10, Line 32, Page 11, Lines 1-2)

(8) In the discussion they mention that “brain tissue reference panels do not have large sample sizes” and how this is a limitation. However, this is also the case for many of the other reference samples, right? And this step (from reference sample to expression imputation) is not covered at all in the current paper. For example, what happens to genetic variants with very low frequency if the reference panel is very small? (For example, if the expression is based on samples smaller than 100 individuals, and the genetic variant is only carried by one or two of the individuals?)

Response: Thanks a lot for your insightful comments! We agree that this limitation is valid for all reference panels with small sample size, not just for brain tissues. Moreover, we did not cover the model training steps in this paper. We had this comment solely based on our finding that larger sample size may lead to higher prediction accuracy in our gene-based PRS analysis, and almost all brain tissues related panels had relatively small sample sizes. We have changed our discussion as follows:

“These limitations may be due to the fact that current brain tissue reference panels, like many other tissues, do not have large sample sizes and/or the associated gene expression imputations may be of low quality. For example, imputations using genetic variants with low frequency may not be accurate when the reference panel sample size is small.”

(Page 14, Lines 13-17)

Reviewers' Comments:

Reviewer #1:

Remarks to the Author:

I appreciate the diligence in addressing the concerns from my first review.

The manuscript has improved significantly at a technical level and added compelling evidence for the result's power via colocalization (fastEnloc).

The GTEx v8 data set adds significant value to the UTMOST results.

However the author's revised analysis introduced two major concerns for me of a technical nature.

Major concerns

A) GTExv8 data was incorporated at the UTMOST analysis.

I'm happy that this was chosen over v7.

However I don't see any benefit to combining v8-based UTMOST results with v6-based UTMOST results.

v6 and v8 share individual samples so that they aren't independent - therefore little gain can be obtained from traditional meta-analysis (even with techniques that factor correlation).

The two criteria for combining v6 and v8 are fundamentally limited by v6's power.

I strongly suggest either explaining the benefit of this approach, or dropping v6-based results.

B) The authors repeatedly say "overlap" (i.e. page 5, line 15) between v6-based and v8-based genes. However GTEx v8 is based on hg38/GRCH38 human genome, using GENCODE v26 for gene annotation

("The GTEx Consortium atlas of genetic regulatory effects across human tissues",

The GTEx Consortium, Science 11 Sep 2020). This is significantly different from GTEx v7 and v6 that are

hg19-based and use older versions of GENCODE.

Therefore significant "coordinate" differences arise between (v6 & v7)-based genes and v8-based genes.

How did the authors reconcile these different genome "coordinates" for coordinate-based notions like overlap?

Did they perform liftover between human genome assemblies?

This must be explicitly described - even if what they did was simply take each GTEx version's gene annotation and take coordinates at face value (in this respect I'm more lax than the community consensus on not mixing coordinates from different assemblies)

Minor concerns

C) fastEnloc's authors (Pividori et al, Science Advances 10 Sep 2020) present it as a very conservative measure,

and use a less stringent threshold (locus RCP > 0.1).

Colloquially, this is interpreted as "presenting evidence of colocalization".

Reporting results with $rcp > 0.5$ and $rcp > 0.9$ is perfectly fine and acceptable: but reporting results with $rcp > 0.1$ might be valuable too, specially for GWAS summary results on traits with less signal. I leave this to the author's discretion.

D) In the rebuttal, the authors mention "European-based LD files provided by fastenloc".

Are they referring to the file currently described in

<https://github.com/xqwen/fastenloc/tree/master/tutorial?>

If so, this is hg38-based whereas typical gwas (UKB, other studies in the author's manuscript) are

hg19-based.

Did they simply match by rsid? This is acceptable but there is some data loss.
If they matched by coordinates, did they reconcile via any means(e.g.liftover)?

E) I disagree with the authors about the remark at page 9, line 26.

- TWAS is appropriate in general as long as it is understood as a statistical measure.

A negative result is interpreted as lack of signal or lack of evidence.

The authors themselves mention this in their discussion (page 13, line 31),

so this remark becomes redundant at best and confusing at worst.

- I don't see the point about discriminating between TWAS signals driven by a single variant or multiple variants.

TWAS is merely a mechanistic approach to infer the effect of a (generally unobserved)

intermediate/molecular trait,

and it is irrelevant whether a single variant tags the mechanism, or many do.

The point of TWAS is actually to detect results that arise from a single variant,

or the evidence combined by multiple weak signals - as the authors themselves remark in their discussion

at page 13, line 21, which is a well-understood property of TWAS.

- From a statistical point of view, TWAS limitations are well understood and colocalization is currently accepted as a good complement ("The GTEx Consortium atlas of genetic regulatory effects across human tissues",

The GTEx Consortium, Science 11 Sep 2020, is merely one such example).

From a point of view of application and treatments,

any TWAS result by itself is insufficient and must be complemented by functional evidence

and generally by experiments as the author themselves acknowledge in discussion.

This makes the author's remark even more debatable.

- In a more general way, the only clarification I think can be made

is that expression is the mechanism assumed, but it is not the only mechanism driving complex traits (i.e. a particular trait might be more influenced by other mechanism such as splicing or methylation).

In other words: a gene's effect on a trait could not be through expression but through splicing

(this is made more evident when interpreting TWAS in the perspective of mendelian randomization).

F) Discussion, page 13, line 32, about causality: this statement is still correct, and I agree that experimental validation is needed for an application, and TWAS can or should be used as a guide for such detailed studies (the same remarks in G) above).

However the addition of colocalization addresses notions of causality, and while still limited by the statistical

nature of such methods, the authors could highlight that they can now discriminate between genes having more evidence

of causal association than others.

H) discussion, page 14, line 9: the better performance is not simply a matter of larger sample size - otherwise using the single tissue with largest sample size would be better than multiple tissues.

Multixcan (Barbeira et al, PLOS genetics Jan 2019) clearly identified that the benefit of integrating multiple tissues

while accounting for the correlation between them effectively amounts to leveraging different expression patterns

accumulated across all tissues, as well as as tissue-specific patterns.

i.e. the gain of multi-tissue approaches is leveraging the biological contexts of both each tissue individually and groups of related tissues.

This has been interpreted elsewhere, from a purely statistical point of view, as a meta-analysis that factors study correlation.

Multi-tissue ASPU (for example Xu et al, Genetics november 201 in the context of image phenotypes

as intermediate phenotypes) provides an alternative study for the benefit of multiple endophenotypes. I suggest the authors change this statement simply to reflect that multi-tissue approaches are superior to single-tissue approaches because they additionally evaluate cross-tissue evidence. (at the risk of being reiterative - if a large sample size is the sole driver of improvement, then methods like UTMOST, Multixcan, omnibus TWAS, or multi-tissue ASPU should not be used at all).

I) In the rebuttal (Part II: Gene-Based PRS with Colocalization):

The authors incorporated colocalization (MOLOC) into their gene-based PRS with a simple but effective technique, which ended up being more than I expected.

I think the authors should consider discussing their technique in their manuscript, although I leave it to their discretion.

The point was to address the most daunting limitation of causality and non-causality from LD, and I think their technique provides enough evidence of their main gene-based PRS analysis to be resilient

to typical confounding factors or distinct causal variants related by LD.

J) Entirely to author's discretion: they mention fastENLOC (page 10, line 20) but reference the previous ENLOC (non-fast) publication. fastENLOC itself is presented and explained in (Pividori et al, Science advances 2020). Is there any reason for this choice?

Reviewer #2:

Remarks to the Author:

Dear authors,

Thank you for addressing all my concerns, I really liked reading this nice paper.

Transcriptome-wide association analysis of brain structures yields insights into pleiotropy with complex neuropsychiatric traits

Responses to the Reviewer 1:

Thank you again for your careful review and constructive suggestions! Here we provide our point-to-point responses. We have made changes in the manuscript accordingly, with changes highlighted. For your convenience, we first restate your comments in italic and then provide our responses.

I appreciate the diligence in addressing the concerns from my first review. The manuscript has improved significantly at a technical level and added compelling evidence for the result's power via colocalization (fastEnloc). The GTEx v8 data set adds significant value to the UTMOST results. However, the author's revised analysis introduced two major concerns for me of a technical nature.

Response: Many thanks for your supportive comments! We have tried our best to address your remaining concerns.

Major concerns:

(1) GTExv8 data was incorporated at the UTMOST analysis. I'm happy that this was chosen over v7. However, I don't see any benefit to combining v8-based UTMOST results with v6-based UTMOST results. v6 and v8 share individual samples so that they aren't independent - therefore little gain can be obtained from traditional meta-analysis (even with techniques that factor correlation). The two criteria for combining v6 and v8 are fundamentally limited by v6's power. I strongly suggest either explaining the benefit of this approach, or dropping v6-based results.

Response: Thank you very much for your comments! We agree that GTEx v8 database has higher power and our approach (combining v6 and v8 results) might be conservative. However, because v6 and v8 share individual samples (as you mentioned), we are more confident to report the associations that can be detected in both v6 and v8 versions. We have added more explanation for our approach in the paper:

"As the GTEx v6 and v8 databases share individual-level samples, we are particularly interested in the associations that can be consistently detected in the two versions. Thus, in the rest of this paper..." (Page5, Line 15)

(2) The authors repeatedly say "overlap" (i.e. page 5, line 15) between v6-based and v8-based genes. However, GTEx v8 is based on hg38/GRCH38 human genome, using GENCODE v26 for gene annotation ("The GTEx Consortium atlas of genetic regulatory effects across human tissues", The GTEx Consortium, Science 11 Sep 2020). This is significantly different from GTEx v7 and v6 that are hg19-based and use older versions of GENCODE. Therefore significant "coordinate" differences arise between (v6 & v7)-based genes and v8-based genes. How did the authors reconcile these different genome "coordinates" for coordinate-based notions like overlap? Did they perform liftover between human genome assemblies? This must be explicitly described - even if what they did was simply take each GTEx version's gene annotation and take coordinates at face value (in this respect I'm more lax than the community consensus on not mixing coordinates from different assemblies)

Response: Thank you very much for your in-depth suggestions! We did take coordinates at face value. Specifically, we performed UTMOST analysis for GTEx v6 and v8 versions separately. After the analysis, we obtained a list of significant gene-level associations for each version. Then we checked whether the same gene (for example, *TREH*) was detected in both lists. If one gene was significant in one version but not in the second version, we further checked whether any of its neighboring genes (within $\pm 1\text{MB}$ window in the second version) were significant. In addition, when training UTMOST models using GTEx data, we used rsid as the SNP reference and used the corresponding gene reference build for each version (GRCh38 for GTEx v8 and GRCh37 for GTEx v6). We have added more descriptions in our main text and supplementary file:

"For each cohort, we obtained a list of significant associations for GTEx v6 and v8 versions, respectively. We reported genes that were either 1) significant in both versions; or 2) significant in one version and at least one of its neighboring (within $\pm 1\text{MB}$ window) gene was significant in the other version." (Page16, Line 3)

"In terms of the cis-SNPs used in imputation models, we used rsid as the SNP reference. For a given tissue, the imputation models for a specific gene were saved as multiple records in the database. Each record corresponded to the weight of an rsid with respect to a gene. As for the range of cis-SNPs for a gene, we used the corresponding gene reference build (GRCh38 for GTEx v8) to identify the cis-SNPs." (Supplementary file, Section 2.3)

Minor concerns:

(3) fastEnloc's authors (Pividori et al, Science Advances 10 Sep 2020) present it as a very conservative measure, and use a less stringent threshold (locus RCP > 0.1). Colloquially, this is interpreted as "presenting evidence of colocalization". Reporting results with $rcp > 0.5$ and $rcp > 0.9$ is perfectly fine and acceptable: but reporting results with $rcp > 0.1$ might be valuable too, specially for GWAS summary results on traits with less signal. I leave this to the author's discretion.

Response: Thank you for letting us know! We have updated our Supplementary Data 14 to report all results with $RCP > 0.1$. We have also updated our main text accordingly:

"We found that 96 of the 278 (34.5%) genes (involving 233 of 918 gene-trait associations) had regional colocalization probability (RCP) > 0.1 in at least one tissue type and seven genes (involving 17 gene-trait associations) had $RCP > 0.9$ (Supplementary Data 14)." (Page10, Line 24)

(4) In the rebuttal, the authors mention "European-based LD files provided by fastenloc". Are they referring to the file currently described in <https://github.com/xqwen/fastenloc/tree/master/tutorial/>?

If so, this is hg38-based whereas typical gwas (UKB, other studies in the author's manuscript) are hg19-based. Did they simply match by rsid? This is acceptable but there is some data loss. If they matched by coordinates, did they reconcile via any means(e.g.liftover)?

Response: Thank you for pointing this out! Yes, the "European-based LD files provided by fastenloc" is the "eur_ld.hg38.bed" file on <https://github.com/xqwen/fastenloc/blob/master/tutorial/>. Yes, we performed liftover in this analysis. Specifically, the coordinates in this file were lifted from hg38 to hg19 using the UCSC Genome Browser (<https://genome.ucsc.edu/cgi-bin/hgLiftOver>).

(5) I disagree with the authors about the remark at page 9, line 26.

- TWAS is appropriate in general as long as it is understood as a statistical measure. A negative result is interpreted as lack of signal or lack of evidence. The authors themselves mention this in their discussion (page 13, line 31), so this remark becomes redundant at best and confusing at worst.

- I don't see the point about discriminating between TWAS signals driven by a single variant or multiple variants. TWAS is merely a mechanistic approach to infer the effect

of a (generally unobserved) intermediate/molecular trait, and it is irrelevant whether a single variant tags the mechanism, or many do. The point of TWAS is actually to detect results that arise from a single variant, or the evidence combined by multiple weak signals - as the authors themselves remark in their discussion at page 13, line 21, which is a well-understood property of TWAS.

- From a statistical point of view, TWAS limitations are well understood and colocalization is currently accepted as a good complement ("The GTEx Consortium atlas of genetic regulatory effects across human tissues", The GTEx Consortium, Science 11 Sep 2020, is merely one such example). From a point of view of application and treatments, any TWAS result by itself is insufficient and must be complemented by functional evidence and generally by experiments as the author themselves. This makes the author's remark even more debatable.

- In a more general way, the only clarification I think can be made is that expression is the mechanism assumed, but it is not the only mechanism driving complex traits (i.e. a particular trait might be more influenced by other mechanism such as splicing or methylation). In other words: a gene's effect on a trait could not be through expression but through splicing (this is made more evident when interpreting TWAS in the perspective of mendelian randomization).

Response: Thank you very much for your suggestions and detailed explanations! We totally agree with you and have removed our remark on page 9.

(6) Discussion, page 13, line 32, about causality: this statement is still correct, and I agree that experimental validation is needed for an application, and TWAS can or should be used as a guide for such detailed studies (the same remarks in G) above).

However, the addition of colocalization addresses notions of causality, and while still limited by the statistical nature of such methods, the authors could highlight that they can now discriminate between genes having more evidence of causal association than others.

Response: Thank you for your suggestions! We have added the following sentence to highlight this point:

"In addition, colocalization analysis (such as fastENLOC) can also help prioritize genes having more evidence of causal association." (Page14, Line3)

(7) Discussion, page 14, line 9: the better performance is not simply a matter of larger sample size - otherwise using the single tissue with largest sample size would be better than multiple tissues. Multixcan (Barbeira et al, PLOS genetics Jan 2019) clearly identified that the benefit of integrating multiple tissues while accounting for the correlation between them effectively amounts to leveraging different expression patterns accumulated across all tissues, as well as as tissue-specific patterns.

i.e. the gain of multi-tissue approaches is leveraging the biological contexts of both each tissue individually and groups of related tissues.

This has been interpreted elsewhere, from a purely statistical point of view, as a meta-analysis that factors study correlation. Multi-tissue ASPU (for example Xu et al, Genetics november 201 in the context of image phenotypes as intermediate phenotypes) provides an alternative study for the benefit of multiple endophenotypes. I suggest the authors change this statement simply to reflect that multi-tissue approaches are superior to single-tissue approaches because they additionally evaluate cross-tissue evidence. (at the risk of being reiterative - if a large sample size is the sole driver of improvement, then methods like UTMOST, Multixcan, omnibus TWAS, or multi-tissue ASPU should not be used at all).

Response: Many thanks for your suggestions and pointing these studies to us! As suggested, we have changed our statement and cited the related references:

“The better performance of cross-tissue analysis may be partially explained by the fact that multi-tissue approaches additionally evaluate cross-tissue evidence^{108,109}.”
(Page14, Line10)

(8) In the rebuttal (Part II: Gene-Based PRS with Colocalization):

The authors incorporated colocalization (MOLOC) into their gene-based PRS with a simple but effective technique, which ended up being more than I expected. I think the authors should consider discussing their technique in their manuscript, although I leave it to their discretion. The point was to address the most daunting limitation of causality and non-causality from LD, and I think their technique provides enough evidence of their main gene-based PRS analysis to be resilient to typical confounding factors or distinct causal variants related by LD.

Response: Thank you for your suggestions! As suggested, we have added and discussed our approach in the supplementary file (Section 2.1.5).

(9) Entirely to author's discretion: they mention fastENLOC (page 10, line 20) but reference the previous ENLOC (non-fast) publication. fastENLOC itself is presented and explained in (Pividori et al, Science advances 2020). Is there any reason for this choice?

Response: Thank you for pointing this out! We have updated the reference for fastENLOC.